

# PIXELCRAFT: A MULTI-AGENT SYSTEM FOR HIGH-FIDELITY VISUAL REASONING ON STRUCTURED IMAGES

**Shuoshuo Zhang**[φπ*] **Zijian Li**[φρ*] **Yizhen Zhang**[φπ] **Jingjing Fu**[φ] **Lei Song**[φ]
**Jiang Bian**[φ] **Jun Zhang**[ρ†] **Yujiu Yang**[π†] **Rui Wang**[φ†]
[π]Tsinghua University    [ρ]Hong Kong University of Science and Technology
[φ]Microsoft Research Asia
{zss24,zhangyizhen24}@mails.tsinghua.edu.cn, zijian.li@connect.ust.hk
{jifu,lesong,jiabia,ruiwa}@microsoft.com
eejzhang@ust.hk, yang.yujiu@sz.tsinghua.edu.cn

## ABSTRACT

Structured images (e.g., charts and geometric diagrams) remain challenging for multimodal large language models (MLLMs), as perceptual slips can cascade into erroneous conclusions. Intermediate visual cues can steer reasoning; however, existing cue-based methods are constrained with low-fidelity image processing and linear, rigid reasoning patterns, limiting their effectiveness on complex structured-image tasks. In this paper, we propose PixelCraft, a novel multi-agent system for high-fidelity image processing and flexible visual reasoning on structured images. The system comprises a dispatcher, a planner, a reasoner, critics, and a set of visual tool agents. To achieve high-fidelity processing, we construct a high-quality corpus and fine-tune an MLLM into a grounding model, whose pixel-level localizations are integrated with traditional computer vision (CV) algorithms in tool agents. Building on this foundation, PixelCraft facilitates flexible visual reasoning through a dynamic three-stage workflow of tool selection, agent discussion, and self-criticism. Moreover, unlike prior linear reasoning patterns that simply append historical images, PixelCraft maintains an image memory to allow the planner to adaptively revisit earlier visual steps, explore alternative reasoning branches, and dynamically adjust the reasoning trajectory during discussion. Extensive experiments on challenging chart and geometry benchmarks demonstrate that Pixel-Craft significantly improves visual reasoning performance for advanced MLLMs, setting a new standard for structured image reasoning. Our code is available at https://github.com/microsoft/PixelCraft.

## 1 INTRODUCTION

Structured images, such as charts and geometric diagrams, pose formidable challenges to current multimodal large language models (MLLMs) (Wang et al., 2025; Yin et al., 2024; Li et al., 2025a; Xia et al., 2025). Whereas natural images are typically characterized by features such as objects, textures, and local visual patterns, which can be effectively captured by existing vision models, structured images encode symbolic and structural elements such as coordinates, data points, line connections, and numerical annotations. Interpreting these structural representations requires precise symbolic abstraction rather than mere pattern recognition. Moreover, reasoning over structured images requires much higher granularity and accuracy. While coarse visual features may suffice for understanding natural images, even subtle differences in structured images (e.g., slightly misreading the height of a single bar) can dramatically alter the interpretation and downstream reasoning.

To enhance the understanding and reasoning ability of structured images, initial strategies, represented by chain of thought (CoT) (Wei et al., 2022), focused on fine-tuning with specialized textual

---

[*]Equal contribution, author order does not indicate contribution. Work done during internship at Microsoft.
[†]Corresponding authors.

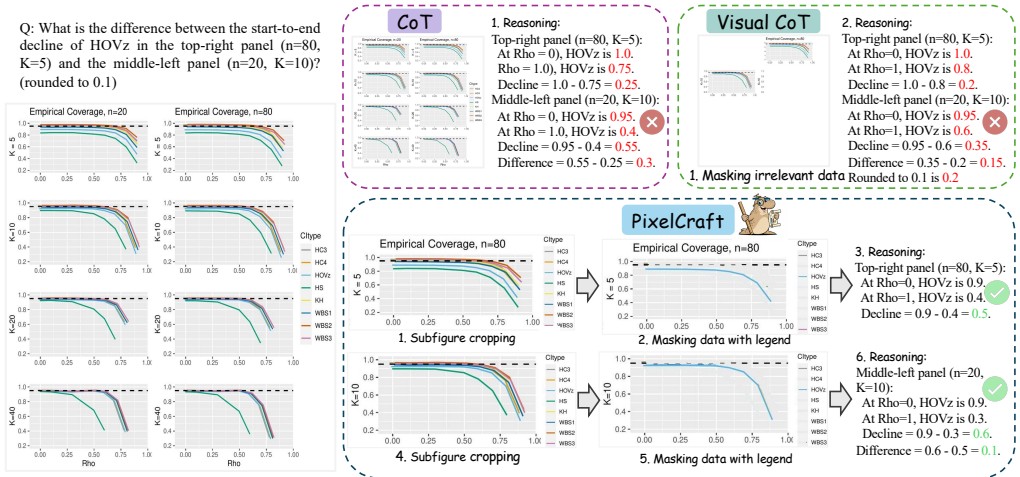

Figure 1: Comparison between CoT, visual CoT, and our proposed PixelCraft. Compared with existing methods, PixelCraft enables high-fidelity image processing and flexible visual reasoning.

reasoning paths (Masry et al., 2024b; Huang et al., 2025; Xu et al., 2024). However, relying solely on textual reasoning often leads to the loss of fine-grained spatial and structural information, making it difficult to capture patterns such as subtle visual differences or geometric constraints. Recent studies have attempted to address this limitation by constructing intermediate visual clues to support visual CoT (Meng et al., 2023; Hu et al., 2024; Fu et al., 2025). Despite this innovation, existing methods either depend on underlying source codes, which is often unavailable, or provide only low-fidelity image processing. Consequently, their applicability is restricted to a specific set of structured images, and their performance remains limited on increasingly complex and realistic benchmarks, such as CharXiv (Wang et al., 2024) and ChartQAPro (Masry et al., 2025).

Besides high-fidelity image processing, structured image understanding also requires accurate and flexible multi-step visual reasoning. However, most existing studies focus on one-step editing or adopt a chain-like linear paradigm, where each intermediate image is derived solely from its predecessor (Hu et al., 2024; Fu et al., 2025; Kumar et al., 2024). This linear approach imposes a cognitive rigidity, forcing the model into a monologue of one-way reasoning. Sophisticated visual analysis, however, is inherently a non-linear process of hypothesis testing, formulating a premise, exploring its implications visually, and, upon encountering contradictions, backtracking to revise earlier assumptions. While a few works on natural images have hinted at non-linearity through zoom-in or multi-region marking (Yang et al., 2023a; Su et al., 2025), they cannot support the rich, recursive exploration needed for structured images. As the example illustrated by Fig. 1, flexible visual reasoning is needed for structured image understanding, which requires the ability to recall historical images, explore multiple reasoning branches, and dynamically adjust the trajectory.

To fill this research gap, we propose **PixelCraft**, a multi-agent system designed for high-fidelity image processing and flexible visual reasoning of structured images. The system comprises a dispatcher, a planner, a reasoner, a planning critic, a visual critic, and a suite of visual tool agents. To dismantle the barrier of low-fidelity processing, we architect a synergistic approach: a compact MLLM (Qwen2.5-VL-3B (Bai et al., 2025)), fine-tuned on our synthetic corpus for precise pixel-level grounding, acts as a "smart eye" to map textual references to coordinates. These coordinates then drive classical computer-vision (CV) operators within our tool agents, which act as "robotic hands" to perform precise image edits. To break free from the cognitive straightjacket of linear reasoning, we propose a three-stage, planner-centric workflow: 1) the dispatcher performs query-aware agent selection; 2) the planner coordinates a role-driven discussion among agents to assemble the visual reasoning process; and 3) a planning critic inspects the trace for errors, triggering re-reasoning. Crucially, we introduce a planner-managed image memory that functions as a "cognitive whiteboard." Instead of indiscriminately feeding all historical images into an ever-expanding context, the planner stores intermediate clues and selectively recalls them for subsequent steps. This enables flexible branching and backtracking, departing from the ephemeral, one-way nature of prior visual CoT methods while simultaneously reducing long-context overhead. The main contributions of our work are summarized as follows:

- We propose PixelCraft, a novel multi-agent system for structured image reasoning that integrates query-aware agent selection, agent discussion, and iterative self-correction with a planner-managed image memory. It features a planner-managed image memory that selectively recalls prior visual states instead of streaming all images, enabling explicit branching and backtracking while keeping context compact. This non-linear, discussion-centric workflow departs from classic visual-interleaved chains and delivers flexible, high-fidelity reasoning on structured images with reduced long-context overhead and fewer visual errors.

- PixelCraft enables high-fidelity, pixel-level image processing to enhance the visual reasoning performance on structured images. We construct a high-quality corpus to finetune a compact MLLM (Qwen2.5-VL-3B) for high-fidelity, pixel-level grounding; its precise coordinates drive classical CV operators inside tool agents, yielding precise, robust and faithful edits.

- PixelCraft shows strong empirical gains on structured-image benchmarks with component-wise evidence. The gains are consistent and significant across widely used and challenging benchmarks (CharXiv, ChartQAPro, EvoChart) and hold across diverse advanced backbones (GPT-4o, GPT-4.1-mini, and Claude 3.7 Sonnet).

## 2 RELATED WORKS

**Visual understanding of structured images.** The visual understanding of structured images like charts remains a significant challenge for multimodal models (Wang et al., 2024; Masry et al., 2025; Xia et al., 2025; Yang et al., 2024). Pioneering works adopted a two-stage approach, using OCR for data extraction followed by LLM reasoning (Liu et al., 2022; Lee et al., 2023). To enhance the reasoning capabilities, subsequent research leveraged Chain of Thought (CoT) data to fine-tune models (Masry et al., 2022; Han et al., 2023; Masry et al., 2024b;a; Meng et al., 2024; Li et al., 2025b), with recent works introducing reinforcement learning (RL) (Chen et al., 2025). Despite these advances, performance remains unsatisfactory on challenging benchmarks, particularly when relying solely on models' intrinsic capabilities.

**Tool use for visual reasoning.** Recent advancements in MLLMs have been significantly propelled by their integration with external tools like OCR and search engines (Liu et al., 2025; Yin et al., 2024; Wang et al., 2025; Li et al., 2025a). Early tool-augmented methods, such as MM-REACT (Yang et al., 2023b) and ViperGPT (Surís et al., 2023), primarily generate textual or code-based outputs, orchestrating calls to vision models or generating Python scripts to solve visual queries. To address more complex tasks, a subsequent line of research has focused on generating intermediate visual evidence. A prominent example, Set-of-Mark (SoM) (Yang et al., 2023a), overlays explicit markers onto image regions to provide the model a clear visual vocabulary, a paradigm successfully extended to tasks like visual grounding (Jia et al., 2024; Wu et al., 2024; Gao et al., 2024; Liu et al., 2024) and object detection (Zhou et al., 2024; Hu et al., 2024). Our work diverges by focusing on visual reasoning for structured images, a domain demanding high-fidelity processing. Recent works Refocus (Fu et al., 2025) and OpenThinkImg (Su et al., 2025) also employ visual tools for chart understanding. However, their toolsets are often specialized—for example, relying on contour or line detection—which limits generalizability. Moreover, their linear, chain-like reasoning is inflexible: it lacks backtracking and branching while overlooking recent findings that MLLM performance degrades on multi-image inputs (Tian et al., 2025; Zhang et al., 2025d).

**Multi-agent collaboration**. The multi-agent collaboration paradigm enhances reasoning beyond single-agent systems. A prominent approach is "multi-agent debate", where agents independently generate solutions and then converge on a final answer through aggregation or discussion, thereby improving reasoning quality (Liang et al., 2023; Du et al., 2023). This principle of leveraging solution diversity is also evident in frameworks for voting (Wang et al., 2022), medical decision-making (Kim et al., 2024), and group discussion (e.g., ReConcile (Chen et al., 2024)). In contrast to these "debate-and-select" methods, a distinct paradigm focuses on structured collaboration among specialized agents (Wu et al., 2023; Li et al., 2023). A popular methodology is role-playing (Shao et al., 2023), where a task is decomposed and collaboratively addressed by agents in specific roles, like a planner or reasoner (Wu et al., 2023). Our work builds upon this role-playing concept by defining a team of agents with synergistic capabilities for planning, image processing, reasoning, and criticizing. While recent work explores dynamic tool generation (Zhang et al., 2025b), creating

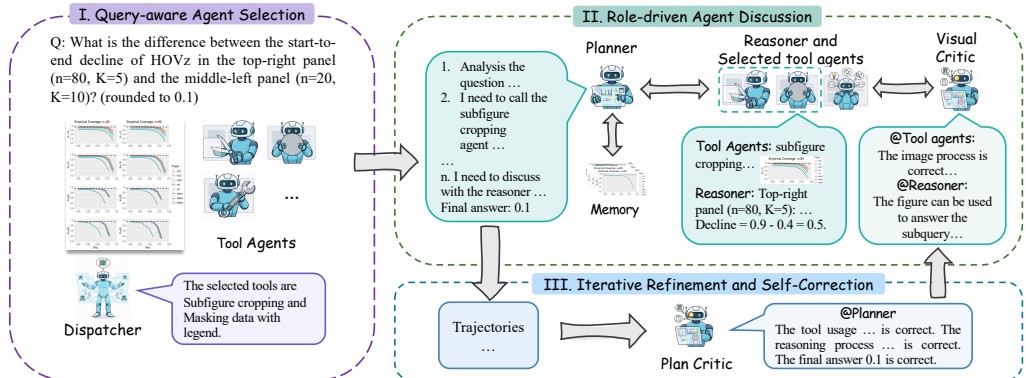

Figure 2: An illustration of the PixelCraft workflow. The process begins with Agent Selection, where the dispatcher chooses the appropriate tools. Next, during Agent Discussion, the planner coordinates tool agents to process the image (e.g., cropping and masking) and the reasoner to perform analysis, with the visual critic providing real-time validation. Finally, the planning critic performs a post-hoc review of the entire process, confirming its correctness.

high-fidelity visual tools automatically remains challenging. Our framework equips dedicated tool agents based on the LLM-generated tools.

## 3 PixelCraft

We consider the complex visual reasoning tasks of chart and geometric understanding, which require step-by-step intermediate visual clues to obtain the final answer. To achieve this, we propose high-fidelity agents (PixelCraft), a multi-modal multi-agent system providing high-fidelity and flexible visual reasoning on structured images. Its design is built on two core principles: synergizing MLLM intelligence with CV algorithms for high-fidelity tool use, and enabling flexible, non-linear reasoning through a planner-centric, discussion-based workflow. As illustrated in Fig. 2, MLLMs serve as the dispatcher, planner, reasoner, and critics, collaborating with specialized tool agents to execute a three-stage process: 1) query-aware agent selection, 2) role-driven agent discussion, and 3) iterative refinement and self-correction.

### 3.1 Agent Roles

**Dispatcher.** The dispatcher serves as the initial entry point. It analyzes the query's requirements and intelligently activates only the most relevant tool agents for the reasoning process, improving the efficiency and response quality.

**Planner.** Functioning as the "conductor" of the agent orchestra, the planner orchestrates the entire reasoning process. It decomposes the complex query into manageable sub-tasks, selects the next agent to act, and manages the flow of image and text information. A critical feature here is the introduction of an image memory. Conventional approaches that feed all historical images into the context suffer from severe long-context overhead and are restricted to a linear, chain-like reasoning pattern. The planner's image memory stores all intermediate visual outputs, allowing it to adaptively recall any historical image. This enables the exploration of alternative reasoning branches and facilitates a more flexible and effective visual reasoning process for complex queries.

**Tool Agents.** To address the nuanced requirements of visual analysis, we initially attempted to prompt LLMs to generate specialized tool agents based on the ArxivQA training set (Li et al., 2024). However, we found that these auto-generated tools were often ineffective, primarily due to the lack of precise grounding coordinates, or were simply invalid, suffering from incorrect code execution and producing erroneous visual outputs. To overcome these challenges, we adopt a two-pronged approach: first, we finetune a grounding model on a meticulously curated dataset (Section 4) to provide accurate pixel-level references. Second, we manually refine the invalid tools to ensure their functionality and correctness. This semi-automated approach, combining LLM-based generation with expert-driven validation, proved crucial for building a reliable toolset. Notably, this data-driven

pipeline significantly reduces the manual burden and reliance on domain expertise compared to manual design. The detailed process of tool generation and refinement is available in Appendix B.

For chart reasoning, we develop four visual tool agents: *1) Subfigure cropping* crops a single subfigure from a multi-chart image using a textual description (e.g., "subplot at row 2, column 1"). *2) Region magnification* zooms into a specified region to highlight local details with $x/y$-axis ticks. *3) Adding auxiliary lines* adds a reference line with $x/y$-axis ticks. *4) Masking Data with Legend:* masks irrelevant data series by identifying the color associated with a specified legend item.

For geometric reasoning, we obtain the tool agents based on symbolic entities (points and lines) and their relations: *1) Point Connection* draws a dashed line segment between two specified points to visualize their geometric relation. *2) Perpendicular Line Construction* constructs a line perpendicular to a reference line and passing through a given point. *3) Parallel Line Construction* constructs a line parallel to a reference line and passing through a given point. We also include the code execution tool to perform numerical computation for the above two tasks.

**Reasoner.** The reasoner functions as the system's dedicated analysis expert. Given input images and a query, it applies logical reasoning to interpret structured data within charts and geometric diagrams. While various sophisticated prompting strategies could be employed, we deliberately use a brief generic instruction to demonstrate the inherent power of our high-fidelity tool agents and the overall framework, rather than attributing performance gains to complex prompt engineering.

**Planning Critic and Visual Critic.** To form a robust, two-layer error correction mechanism, we introduce two distinct critics operating at different stages of the workflow: 1) A visual critic for in-loop verification: Visual tools, unlike deterministic coding, can introduce errors. To ensure the fidelity of the reasoning chain, the visual critic operates in-loop goal satisfaction of the processed image (e.g., verifying a crop was successful) and answerability of an image before it is passed to the reasoner. (2) A planning critic for post-hoc refinement: It scrutinizes the sequence of tool usage and logical steps for inefficiencies or errors, such as using a suboptimal tool or a flawed reasoning path. Its feedback is then used to guide the self-correction stage.

## 3.2 WORKFLOW OF PIXELCRAFT

The workflow of PixelCraft is query-aware, role-driven, and visual-critic, which enables high-fidelity image processing and flexible visual reasoning. The system's workflow unfolds across the following three stages:

**Query-aware Agent Selection.** The workflow begins with the dispatcher, which analyzes the incoming query to determine the necessary modalities and processing requirements. Based on this initial triage, the dispatcher selects a relevant subset of tool agents to perform visual reasoning, together with the reasoner. This careful selection ensures that only the most pertinent agents are activated, thereby optimizing both computational efficiency and the relevance of the subsequent process.

**Role-driven Agent Discussion.** Following agent selection, the planner orchestrates the reasoning process. It decomposes the primary query into manageable subqueries, sequences agent activations, and coordinates all inter-agent communication. At each step, the planner dynamically activates the appropriate agent (either a tool agent or the reasoner) and selects an image from the image memory (which contains all historical visual clues and their corresponding descriptions). To ensure high-fidelity execution, the planner assigns a specific goal to tool agents (e.g., "crop the subfigure at row 1 column 1") or poses a precise subquery to the reasoner. Throughout this process, both processed images and textual information—such as goals, subqueries, and analytical outputs—are flexibly exchanged among agents via the planner, enabling each subquery to be tackled with the most suitable capabilities.

To prevent visual errors from propagating, we introduce critical inspection steps. After a tool agent processes an image, the visual critic evaluates its goal satisfaction, i.e., whether the visual output successfully fulfills the goal assigned by the planner. Furthermore, when a processed image and its associated subquery are sent to the reasoner, the visual critic assesses the image's answerability. When an error is detected, the error alert will be returned to the planner for subsequent reasoning, ensuring the reasoning's robustness.

**Iterative Refinement and Self-Correction.** Once an initial answer is generated, the planning critic performs a final review of the entire reasoning process, scrutinizing each step for accuracy, logical consistency, and completeness. By identifying misused tools or erroneous conclusions, the planning critic proposes corrections to the tool list (e.g., adding or removing tools) and provides valuable suggestions to refine the reasoning process, such as proposing better subqueries or tool usage strategies. This feedback serves as additional input for a second attempt where PixelCraft re-answers the query. This procedure enables a cycle of self-improvement, enhancing the system's decision-making and tool selection over time. The self-correction example can be found in Appendix D.1.

Through this structured and collaborative workflow, PixelCraft integrates flexible tool selection, adaptive image memory, robust visual verification, and a self-correcting reasoning process. This synergy achieves accurate, flexible, and interpretable visual reasoning for complex queries.

## 4 GROUNDING FOR HIGH-FIDELITY IMAGE EDITING

Precise visual grounding is essential for downstream image editing tools, such as extracting a specific subplot for further analysis. However, grounding elements within structured visual inputs like charts, scientific plots, and geometric diagrams poses unique challenges due to their abstract and compositional layouts. While the prior work Refocus (Fu et al., 2025) has approached this by locating objects based on specific visual primitives (e.g., lines, squares), its reliance on these predefined structures limits its applicability to a narrow range of chart types. To overcome these limitations, we introduce a hybrid dataset of synthesized charts and geometric diagrams, leveraging it to fine-tune Qwen2.5-VL-3B (Bai et al., 2025) for high-fidelity grounding in these challenging domains. We curate a hybrid dataset from two complementary sources: (1) programmatically synthesized charts and (2) annotated geometric diagrams.

**Programmatically Synthesized Charts.** To generate a large-scale dataset with diverse layouts and precise ground-truth annotations, we propose a multi-stage synthesis pipeline. The initial stage focuses on single-panel charts and involves two primary steps: content specification and diversified rendering. First, we employ GPT-4o (Hurst et al., 2024) to produce structured JSON objects that specify textual elements (e.g., titles, legends, axis labels). Second, we build upon a set of predefined base code templates, which are then programmatically augmented and rewritten by GPT-4o to enhance visual and structural diversity. These modified templates are subsequently rendered into images via Matplotlib. The rendering process is instrumented to extract the precise position coordinates of all visual elements, thereby generating accurate ground-truth for grounding tasks.

To introduce greater structural complexity and align with common subfigure grounding tasks, the second stage of our pipeline composes multi-panel chart figures. We randomly sample, duplicate, and arrange the generated single-panel charts into complex layouts, creating composite images that feature between 2 and 16 subfigures with randomized spatial margins. In total, our pipeline yielded 53k ground-truth annotation pairs, with 43k extracted from our synthesized single-panel charts and the remaining 10k from multi-panel compositions.

**Geometry Annotation.** To enhance our model's geometric grounding capabilities, we augment our dataset with 2,000 samples from the Inter-GPS benchmark (Lu et al., 2021). Each sample in this benchmark consists of a geometric diagram accompanied by rich annotations of its constituent elements and properties. We process these source annotations to extract the position coordinates of geometric points along with their corresponding textual labels. This curated data is specifically utilized for point-level geometric tools.

**Supervised Fine-Tuning.** We formulate structured image grounding as an autoregressive sequence prediction task. Specifically, given an input image $I$ and textual prompt $P$, the model generates a sequence $Y = (y_1, \ldots, y_T)$ that jointly encodes the textual answer and the corresponding bounding boxes. To this end, we finetune Qwen2.5-VL-3B (Bai et al., 2025) with our curated dataset, where spatial locations are represented by absolute coordinates to align with the model's native grounding format. Full training details are provided in Appendix C.1.

The finetuned grounding model provides accurate grounding coordinates with the elements in structured images, enabling high-fidelity image processing for visual reasoning. The quantitative evaluation and example comparison of the model's grounding accuracy are provided in Section 5.3.

Table 1: Performance comparison across different models and evaluation methods

| Method | GPT-4o | | | GPT-4.1-mini | | | Claude-3.7-sonnet | | |
|---|---|---|---|---|---|---|---|---|---|
| | CharXiv | ChartQAPro | EvoChart | CharXiv | ChartQAPro | EvoChart | CharXiv | ChartQAPro | EvoChart |
| Direct answer | 49.6 | 52.51 | 62.64 | 58.6 | 57.85 | 71.28 | 67.1 | 62.83 | 74.16 |
| CoT | 51.1 | 56.52 | 68.64 | 63.8 | 62.21 | 76.64 | 68.3 | 66.07 | 77.92 |
| Debate | 50.7 | 49.38 | 65.52 | 62.4 | 57.75 | 72.72 | 67.7 | 66.58 | 78.08 |
| Reconcile | 52.4 | 54.36 | 65.20 | 63.5 | 59.49 | 72.64 | 68.5 | 65.97 | 78.56 |
| Refocus | 47.2 | 46.30 | 50.88 | 60.7 | 57.24 | 58.96 | 62.4 | 58.42 | 77.06 |
| **Ours** | **55.2** | **58.83** | **70.24** | **68.1** | **65.56** | **79.44** | **73.9** | **69.82** | **80.48** |
| Δ | **+5.6** | **+6.32** | **+7.60** | **+9.5** | **+7.71** | **+8.16** | **+6.8** | **+6.99** | **+6.32** |

Table 2: Accuracy on the *Geometry3K* (Lu et al., 2021) auxiliary-line subset.

| Model | Direct Answer | CoT | Debate | Reconcile | Ours |
|---|---|---|---|---|---|
| GPT-4o | 21.09 | 17.19 | 25.00 | **26.56** | **26.56** |
| GPT-4.1-mini | 24.22 | 26.56 | 29.69 | 32.81 | **34.38** |
| Claude-3.7-sonnet | 30.47 | 28.13 | 28.13 | 32.03 | **33.59** |

Table 3: Results comparing visual CoT paradigm with ours framework.

| | CharXiv | ChartQAPro |
|---|---|---|
| Visual CoT | 65.0 | 61.04 |
| Ours | **68.1** | **65.56** |

## 5 EXPERIMENTS

In this section, we first evaluate PixelCraft on chart and geometric reasoning tasks. Ablation studies demonstrate the effectiveness of each component of PixelCraft. Furthermore, the analysis and discussion demonstrate why PixelCraft improves visual reasoning capabilities in the aspects of grounding accuracy, high-fidelity image processing, flexible visual reasoning, tool usage, and self-correction.

### 5.1 RESULTS ON CHART REASONING

**Benchmarks and Models.** Our evaluation is conducted on three recent and challenging chart reasoning benchmarks: CharXiv (Wang et al., 2024), ChartQAPro (Masry et al., 2025), and EvoChart (Huang et al., 2025). We utilize the reasoning-focused question set for CharXiv as commonly used and the full test sets for both ChartQAPro and EvoChart. Following the protocol from CharXiv (Wang et al., 2024), we employ an LLM-as-a-judge approach using GPT-4.1-mini-20250414 to evaluate the correctness of the final answers. To ensure a fair comparison across all methods, particularly given the long outputs generated by agentic and CoT approaches, we omit the 5% margin typically used for ChartQAPro and apply this unified evaluation standard consistently. We demonstrate the effectiveness of PixelCraft by integrating it with several powerful MLLMs, including GPT-4o-20240806, GPT-4.1-mini-20250414, and Claude-3.7-sonnet.

**Baselines.** We take Direct answer and chain of thought (CoT) methods as the basic baselines. Additionally, we include more advanced baselines in terms of tool using and multi-agent collaboration: *Tool-using method*: Although tool-augmented visual reasoning has been widely explored, chart-specific tools remain limited. To our knowledge, Refocus (Fu et al., 2025) is a representative chart-tool approach, and we use it as our baseline. *Multi-agent methods*: we select two popular multi-agent methods: Debate (Du et al., 2023) and Reconcile (Chen et al., 2024). We set up Debate and Reconcile with two reasoners corresponding to the two agents (planner and reasoner) in our work, where these two agents discuss to converge on a final answer.

**Main results.** Table 1 shows that our proposed PixelCraft significantly outperforms all the baselines on these three chart-reasoning benchmarks across GPT-4o, GPT-4.1-mini and Claude-3.7-Sonnet. Moving from the naive "Direct answer" setting to chain-of-thought (CoT) prompting already provides a boost (nearly 3–6 %), confirming the general value of explicit reasoning. The multi-agent collaboration methods, including Debate and Reconcile, show little benefit, indicating that multiple agents without specialized visual tools struggle to perform visual reasoning on charts. Refocus couples an LLM with chart-related tools, performs inconsistently and in several cases lags even the CoT baseline, suggesting that its visual tools are insufficient for complex structured image processing. In contrast, PixelCraft, equipped with high-fidelity tools powered by our fine-tuned grounding model, consistently achieves the highest accuracy across all benchmarks, demonstrating the clear superiority of our approach.

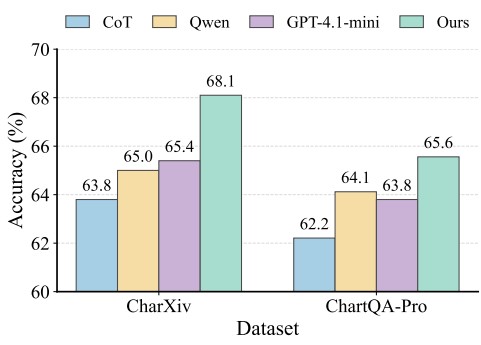 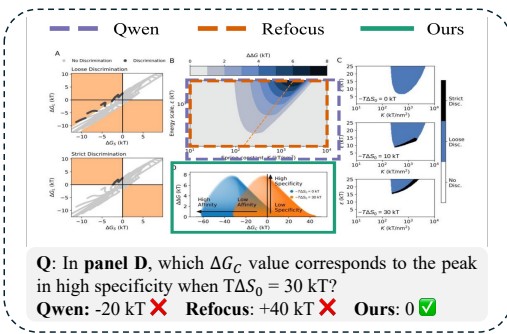

(a) Comparison of grounding methods.   (b) Illustrative grounding example.

Figure 3: Grounding case and method comparison. (a) Comparison of different grounding methods within our agent framework, including our fine-tuned model, Qwen2.5-VL-7B and GPT-4.1-mini. (b) Grounding case showing the performance of Qwen2.5-VL-7B, Refocus and Ours, with colored boxes indicating grounded regions.

## 5.2 RESULTS ON GEOMETRIC REASONING

Complex geometric reasoning often requires the visual clues (i.e., auxiliary lines) to solve, where the auxiliary lines need to be sketched in the geometric image to help visual reasoning. To construct such a specific geometric benchmark with requirements of visual clues, we filter 128 complex test samples that require the intermediate visual clues to answer from the test set of *Geometry3K* (Lu et al., 2021). Since the tools from Refocus (Fu et al., 2025) and Visual Sketchpad (Hu et al., 2024) do not support geometric reasoning tasks or require the source code to perform image processing, which is commonly unavailable, we omit them in the evaluation.

**Main results.** The results in Table 2 demonstrate a clear hierarchy of performance among different reasoning strategies on the auxiliary-line subset of Geometry3K. The effectiveness of baseline methods progresses logically. Compared to a simple "Direct Answer", incorporating a CoT proves beneficial for most models, underscoring the value of explicit reasoning. Furthermore, the multi-agent strategies, Debate and Reconcile, consistently outperform these single-agent approaches. This highlights the power of collaborative frameworks to verify logic and reduce errors in complex tasks. Building upon these insights, our proposed PixelCraft method consistently surpasses all baselines across every model. It achieves the highest accuracy in all test cases, establishing a new state-of-the-art on this challenging benchmark. This consistent and significant advantage validates the superior design and robustness of our method for solving complex geometric problems. Detailed case studies and visual illustrations are provided in Appendix D.2.

## 5.3 ANALYSIS AND DISCUSSION

To further demonstrate the effectiveness of PixelCraft, we first conduct ablation studies for each component and then explore the question *'Why can PixelCraft improve the performance?'*. Additionally, we extend our evaluation with a comprehensive analysis of computational efficiency and generalization to real-world infographics in Appendix E and Appendix F, respectively.

**Ablation studies.** To evaluate the effectiveness of each component of our agent, we conduct a role-wise ablation study. As shown in Table 4, introducing the Tool Agents (TA) provides the largest average performance gain relative to the no-component CoT baseline, underscoring the critical need for specialized tools in visual analysis. Adding the Dispatcher (Disp) provides further improvements, as filtering irrelevant tools leads to more accurate tool usage. The Visual Critic (VC) also contributes positively by filtering invalid processed images and avoiding erroneous answers. When the three are combined, their synergistic effect brings performance to 67.5% on CharXiv and 64.89% on ChartQAPro. Finally, incorporating the Planning Critic (PC) yields the best overall results of 68.1% and 65.56%, respectively. This systematic analysis validates that each component contributes positively and cumulatively to the agent's overall reasoning capabilities.

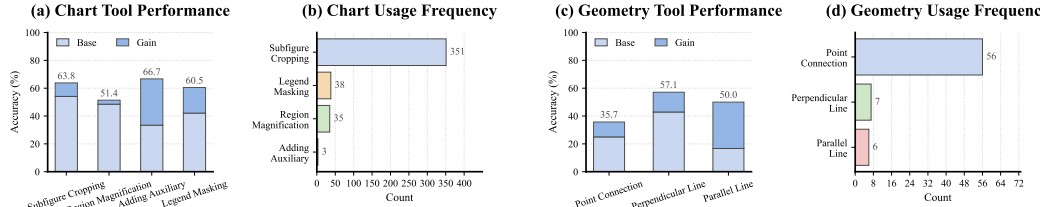

Figure 4: Effectiveness and prevalence of tool usage with GPT-4.1-mini on CharXiv (Wang et al., 2024) and Geometry3K (Lu et al., 2021). The performance gain is calculated specifically on the *subset* of queries where each tool was activated.

| TA | Disp | VC | PC | CharXiv | ChartQAPro |
|----|------|-----|-----|---------|-----------|
| - | - | - | - | 63.8 | 62.21 |
| ✓ | ✗ | ✗ | ✗ | 65.0 | 63.66 |
| ✓ | ✓ | ✗ | ✗ | 65.9 | 64.43 |
| ✓ | ✗ | ✓ | ✗ | 66.0 | 63.96 |
| ✓ | ✓ | ✓ | ✗ | 67.5 | 64.89 |
| ✓ | ✓ | ✓ | ✓ | **68.1** | **65.56** |

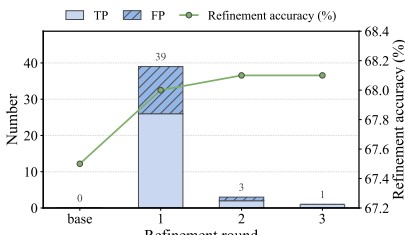

Table 4: Role-wise ablation over Tool Agents (TA), Dispatcher (Disp), Visual Critic (VC), and Planning Critic (PC).

Figure 5: Critic-based erroneous query identification (TP/FP counts) and post-refinement accuracy.

**Comparison with visual CoT.** To validate the architectural benefits of our agent framework, we conducted a controlled comparison against a simplified Visual Chain-of-Thought (CoT) baseline, which simply includes all the historical images for reasoning. This baseline utilized the exact same prompts and tools but employed a single planner to handle both task planning and visual reasoning monolithically. As shown in Table 3, our full agent framework significantly outperforms this simplified approach. This result underscores the superiority of our design of the setting of image memory and image selection beyond the visual tools, which allows flexible visual reasoning by allowing image selection and image recall for alternative branches of reasoning.

**High-fidelity image edits benefit visual reasoning.** Precise localization of chart components is a critical prerequisite for accurate data extraction, directly contributing to superior performance on downstream chart analysis benchmarks. To achieve this, we finetuned a grounding model that substantially boosts localization accuracy, increasing the overall IoU from 0.26 to 0.93 compared to the base model (details in Appendix D.3). The impact of this enhancement is validated on our downstream benchmark. As shown in Fig. 3a, our proposed agent system with the finetuned grounding model significantly outperforms that with the base model. Fig. 3b provides a visual explanation for this performance gap: our model accurately localizes the queried subfigure while both the base model and the Refocus method fail, thus enabling more precise subsequent processing.

**Analysis on visual tool usage frequency and performance gain.** We analyze the tool usage frequency and the performance gain of each visual tool. As shown in Fig. 4, LLMs achieve an imbalanced tool calling frequency on CharXiv and Geometry3K, with more preference on Subfigure Cropping and Point Connection. This is because the tool calling is query-driven and image-driven. Most of the images on CharXiv are multi-chart images, with queries about deep analysis on a single subfigure or comparison between subfigures. The queries on Geometry3K mainly require point connection for mathematical analysis. All of the visual tools improve the performance over the base CoT method without visual tools, with a significant improvement on some tools. For instance, Making Data with Legend improves 18.4% accuracy on CharXiv and Parallel Line Construction holds an accuracy improvement of 50% on Geometry3K. This demonstrates the necessity and effectiveness of our proposed visual tool agents.

**PixelCraft can identify and refine the erroneous answering.** After the answer generation, PixelCraft will review the tool usage and the whole reasoning process to identify the erroneous answers for the next-round reanswering. We conduct three-round reviewing and reanswering to evaluate the identification and the reanswering accuracy. Fig. 5 shows that most of the identified queries are

incorrect-answering (true positive), and the identification number significantly drops to near 0 after 2 rounds. Furthermore, after reanswering with the updated tool and suggestions, PixelCraft refines some incorrect answers, resulting in higher accuracy.

## 6 CONCLUSION

In this paper, we proposed PixelCraft, a novel multi-agent system that unifies high-fidelity, pixel-level tool agents with a flexible reasoning workflow for structured image reasoning. Tool agents, powered by a fine-tuned pixel-level grounding model and classical CV operators, deliver precise image edits that preserve critical visual evidence. Building on these trusted tool agents, coordinated planner, reasoner, and critic workflows, supported by image memory, enable non-linear, multi-branch exploration and targeted re-reasoning. Across challenging chart and geometry benchmarks, PixelCraft consistently elevates strong MLLMs, establishing a new standard for reliable and sophisticated multimodal reasoning.

## ACKNOWLEDGEMENTS

This work was partly supported by the National Natural Science Foundation of China (Grant No. 62576191) and the Shenzhen Science and Technology Program (ZDCY20250901103533010).

## ETHICS STATEMENT

This work adheres to the ICLR Code of Ethics. No human subjects or animal experiments were involved in this study. All datasets and language models employed are publicly available, widely adopted in the research community, and contain no personally identifiable information (PII) or sensitive content. We have taken steps to identify and mitigate potential biases in data selection, model prompting, system design, and evaluation to ensure fairness and avoid harmful or discriminatory outcomes. The agent system is developed and evaluated solely for research purposes, with safeguards in place to minimize misuse and unintended consequences. No personal identities were collected, used, or disclosed in this research. All procedures comply with relevant legal, privacy, and data protection requirements, and the study was conducted in accordance with established research ethics standards.

## REPRODUCIBILITY STATEMENT

To ensure the reproducibility of our work, we provide detailed information regarding our models, prompts, and code. Appendix C of this paper includes the training details for our grounding model and the core prompts utilized by the agent framework. Furthermore, the source code for our framework is accessible at the following anonymous repository: `https://anonymous.4open.science/r/PixelCraft-DB52`.

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

## A   USE OF LARGE LANGUAGE MODELS

In adherence to ICLR 2026 policy, we disclose the assistive use of Large Language Models (LLMs) in the preparation of this paper. The models were employed to refine grammar and improve the clarity of the text. Furthermore, an LLM was utilized to synthetically generate a portion of our dataset and tool codes. The authors have reviewed all LLM-generated contributions and take full responsibility for the content and integrity of this work.

## B   AUTOMATED GENERATION OF VISUAL TOOLS

To automate the discovery and creation of visual tools, we leveraged GPT-4.1-mini to programmatically generate potential tool candidates. We initiated this process by first randomly sampling 500 questions from ArxivQA (Li et al., 2024), which comprises a diverse set of charts from arXiv papers. Following this, we engineered a detailed prompt designed to instruct the model to generate visual tool codes and corresponding tool descriptions.

In total, this process generated 468 tool candidates. To systematically organize the output, we first encoded the name and description of each tool into a vector representation using Qwen3-Embedding-8B (Zhang et al., 2025c). We then performed a clustering analysis on these embeddings to consolidate duplicates and identify key functional categories. An illustrative sample of the resulting clusters is provided in Fig. 6a and Table 5. Our analysis of these clusters revealed that the vast majority of the generated tools fell into a few recurring functional categories, primarily those demanding strong visual grounding capabilities like $Subfigure\_Cropping$ and $Adding\_Auxiliary\_Lines$. Based on this finding, we consolidated the most promising tools from these recurring categories. To improve their robustness and generality, these selected tools were first processed by the GPT-o3 for automated rewriting, followed by a final phase of manual tuning to ensure their correctness and practical usability. This curated set of high-fidelity visual tools formed the core of our agent's toolkit.

Table 5: Cluster statistics and representative tool descriptions.

| Cluster ID | Cluster Size | Tool Definition (examples) | Summary Tools |
|---|---|---|---|
| 0 | 148 | `draw_vertical_guide_line`
`draw_horizontal_guide_line`
`mark_threshold` | Adding Auxiliary Lines |
| 1 | 128 | `highlight_subfigure`
`crop_subfigure`
`isolate_subplot` | Subfigure Cropping |
| 2 | 101 | `highlight_series_mask`
`highlight_mp_curve`
`highlight_data_points` | Masking Data with Legend |
| 3 | 64 | `magnify_region_around_x`
`crop_and_magnify_region`
`crop_region` | Region Magnification |
| *Others* | 27 | `find_peaks_in_data`
`highlight_max_point`
`crop_beta_vs_time_panel`
`highlight_intersection_point`
`draw_spectral_type_guides` | Useless tools |

To probe the breadth of GPT-4.1-mini's generative capabilities beyond this initial set, we conducted a second experiment where we explicitly constrained the model from producing the known grounding-based tools. This constraint prompted a shift towards generating significantly more complex tools, whose procedures were heavily reliant on classical computer vision primitives for precise object detection and data extraction, with representative examples including detecting all data points in a scatter plot to compute their mean, or locating the intersection point of two plotted lines. However, our empirical evaluation revealed that these novel computational tools consistently exhibited low accuracy, even after being subjected to the same GPT-o3 rewriting process (as depicted in Fig. 6b).

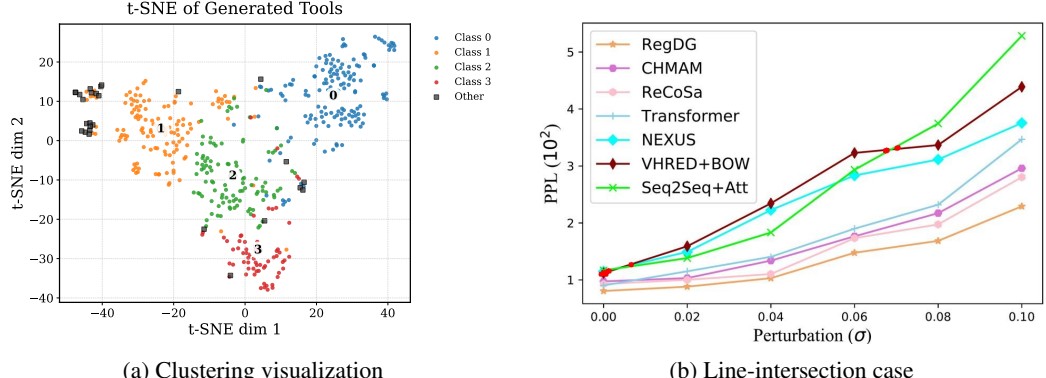

(a) Clustering visualization          (b) Line-intersection case

Figure 6: Left: a t-SNE projection of the generated tool embeddings. Right: a representative case in which classical computer-vision utilities detect the intersection of two lines (red dots) on a chart.

Given the critical impact of tool fidelity on the agent's downstream reasoning performance—and the high risk of error propagation from unreliable tools—we made the strategic decision to exclude this latter set from our agent's final toolkit during evaluation.

## C  IMPLEMENTATION DETAILS

### C.1  TRAINING DETAILS

We performed full-parameter supervised fine-tuning on Qwen2.5-VL-3B-Instruct (Bai et al., 2025) with our custom-collected dataset of 53,000 samples. The model was trained for a single epoch on four NVIDIA A100 GPUs using a learning rate of $1 \times 10^{-5}$. A representative training prompt is shown below:

---

**Example Prompt**

- **System:** You are a helpful assistant specialized in chart analysis. Examine the provided chart image and return the pixel coordinates of the requested element, or return "Not found" if the element does not exist in the image.
- **User:** (`example.png`) This is a chart image. Please locate the pixel coordinates of `<|object_ref_start|>`the subfigure at row 2, column 2`<|object_ref_end|>` in the image.
- **Assistant:** `<|box_start|>[x1, y1, x2, y2]<|box_end|>`

---

### C.2  VISUAL TOOL IMPLEMENTATION DETAILS

This section provides a more detailed breakdown of the implementation for the visual tools introduced in Section 3.1. A core component across these tools is our specialized grounding model, which localizes the elements specified in the planner's query into precise pixel coordinates within the image.. The operational flow for each tool is as follows:

- **Subfigure Cropping:** This tool first employs our grounding model to localize the bounding boxes of both the target subfigure and any relevant legends (e.g., "the legend for plot A") based on the textual description. If multiple elements are identified, such as a chart and its corresponding but spatially separate legend, the tool programmatically crops these regions and composes them into a single, coherent image for analysis.

- **Region Magnification:** For this operation, the planner provides the start and end tick values on the $x$ and $y$ axes that define the region of interest. The grounding model is then tasked with identifying the precise pixel coordinates corresponding to these tick marks.

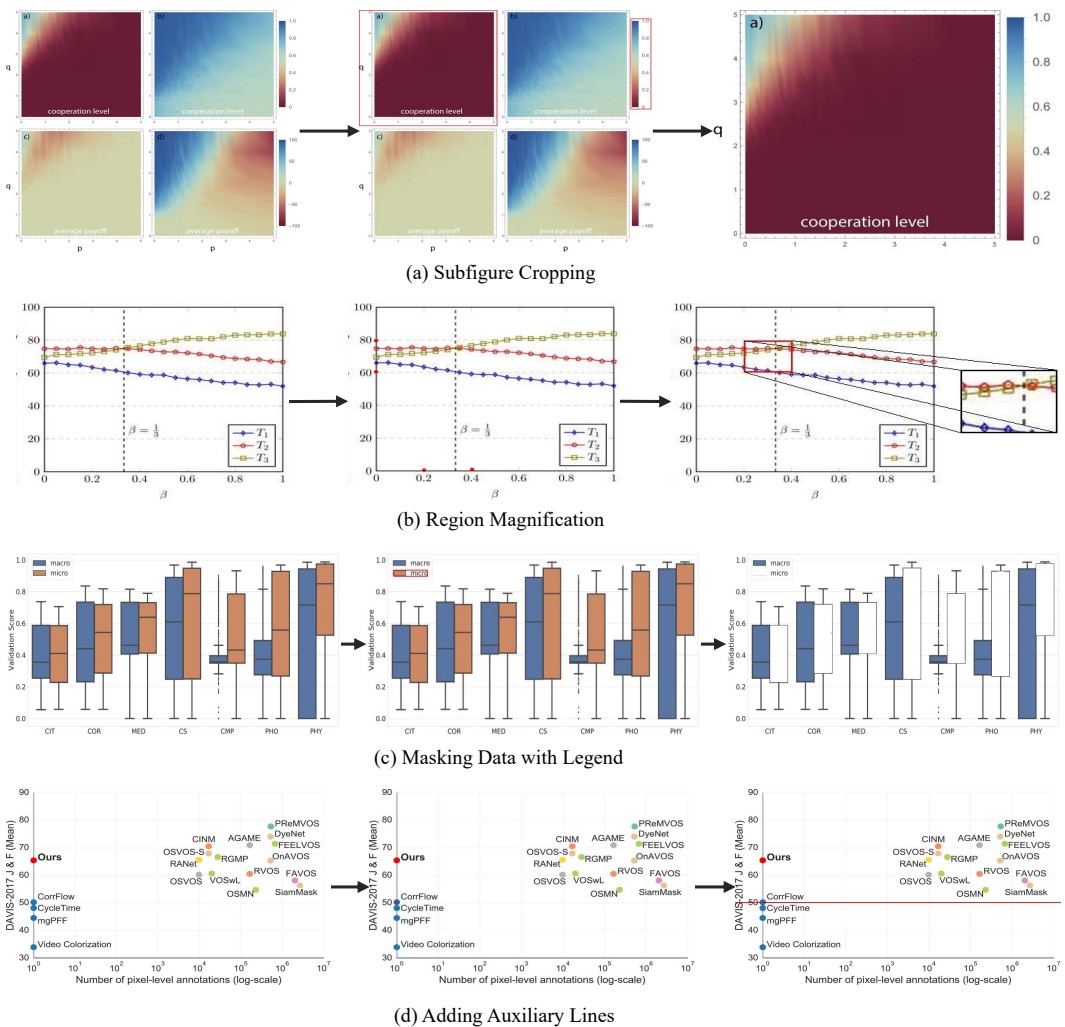

Figure 7: Examples of our visual tools for structured images: (a) cropping subfigures, (b) magnifying regions, (c) masking elements by legend, and (d) adding auxiliary lines.

Subsequently, the tool performs a crop operation based on these coordinates and resizes the resulting region to achieve magnification, preserving local details.

- **Adding Auxiliary Lines:** Similarly to region magnification, this tool leverages the grounding model to determine the pixel locations of specified tick values on an axis. It then renders a horizontal or vertical line at this position to serve as a visual reference for comparison or thresholding tasks.

- **Masking Data with Legend:** This tool operates in a two-stage process. First, the grounding model localizes the specified legend item (e.g., "the line representing Group A") within the chart. Second, the tool extracts the dominant color from the localized legend's icon. This extracted color value is then used to create a binary mask for the corresponding data series in the plot area, effectively isolating it from other visual elements.

Furthermore, our Geometric Tools share a unified implementation pipeline where the grounding model is first invoked to detect the key points (e.g., line endings, vertices) specified by the query. The tool then programmatically renders the desired line based on these geometric constraints. Illustrative case studies for these tool implementations are presented in Fig. 7.

## C.3 PROMPT TEMPLATES

---

**Planner**

You are an expert on chart understanding to analyze the question step by step until answering the final answer. You are given the chart figure `<|image|>` with the image path being `<|image_path|>`, and need to solve the following question: `<|question|>`. The question can be solved via the intermediate processed image using tools. To solve the complex task, you can decompose it into some simple subquestions and use tools to generate intermediate reasoning processes until getting the final answer.

**Available Tools**
Use the following tools to analyze and process images:
`<|tool_descriptions|>`
**Image Pool** `<|image_pool|>`
**Instructions**

1. **Workflow:**

    - Actions should be logical and tool-driven rather than directly outputting the answer.
    - Each THOUGHT analyzes progress; if the final answer is reached, output it in `FINAL ANSWER` with `TERMINATE` in the **ACTION** section.
    - Select tools based on their descriptions and constraints; if one fails, try alternatives.
    - Always use image paths from the image pool for tool inputs.

2. **Final Answer:** Provide a concise summary when the solution is found, then end with `TERMINATE`.

3. **Format:** Follow the structure strictly:

    ```
    THOUGHT N: [Analysis]
    ACTION N: tool_name(key=value)
    OBSERVATION N: [Result]
    ```

**Response History:** `<|history|>`
**Current Step Template:**

```
THOUGHT {i}: [Your analysis]
ACTION {i}: tool_name(key=value)
OBSERVATION {i}: [Result or observation]
```

---

**Reasoner**

- **System:** You are an expert in chart analysis and data visualization. Please analyze the chart carefully and provide accurate answers based on the visual data presented.
- **User:** (`example.png`) {question}

---

**Visual Critic**

Please check whether the given image contains enough information to answer the question: `<|question|>`.
**Examples:**

- If the question requires value extraction from a chart, the image should contain the chart with clear axis and data points.
- If the question is about the trend of a chart, it need not contain the chart title or axis labels.
- If the question is about one subchart compared to another, it is acceptable to have only one subchart in the image.

- If the image is extremely incomplete or the question is not related to the image, return `false`.

- If the image is complete and the question is suitable for answering with the image, return `true`.

---

**Planning Critic**

You are a helpful AI assistant. You are given a question and a chart image. You need to reason with the question and the chart image, and then evolve the plan to answer the question by adjusting the tools and the corresponding parameters. If the current plan is perfect, you should maintain the current plan.
**Available Tools:** `<|tool_descriptions|>`
**Question:** `<|question|>`
**Reasoning Process:** `<|current_plan|>`
**Instructions:**

1. Analyze the question and the answer to assess the correctness of the final answer based on the image.

2. Some intermediate outputs from tools may be incorrect; analyze the correctness of the extracted information.

3. If the final answer is correct, mark `ADJUSTMENT: False`; otherwise, mark `ADJUSTMENT: True`.

4. If the final answer is incorrect, analyze the flawed reasoning process and extracted information, with `ADJUSTMENT: True`.

5. If some tools failed or were incorrectly applied during the reasoning process, remove them from the tool list. Output the updated tool list as:

   ```
   tools: [tool1, tool2, tool3]
   ```

6. If `ADJUSTMENT: True`, provide detailed suggestions for the incorrect parts.

---

**Dispatcher**

You are an expert chart analyst specializing in automated tool selection for chart analysis tasks.
**Task Overview** Given a chart image and a related question, you must analyze both inputs and select all possible tools from the list that can help answer the question.
**Available Tools** `<|tool_descriptions|>`
**Selection Guidelines**

1. **Priority Order:** Consider tools in order of their relevance to the question type.

2. **Empty Selection:** Return `[]` if no tools are suitable.

3. **Context Awareness:** Consider the description and constraints of each tool carefully.

**Analysis Framework**

1. **Question Analysis:** Identify the core task (counting, comparison, extraction, calculation).

2. **Chart Assessment:** Determine chart structure (single/multi-chart, complexity).

3. **Tool Matching:** Map identified needs to available tools.

4. **Validation:** Ensure selected tools satisfy constraints and requirements.

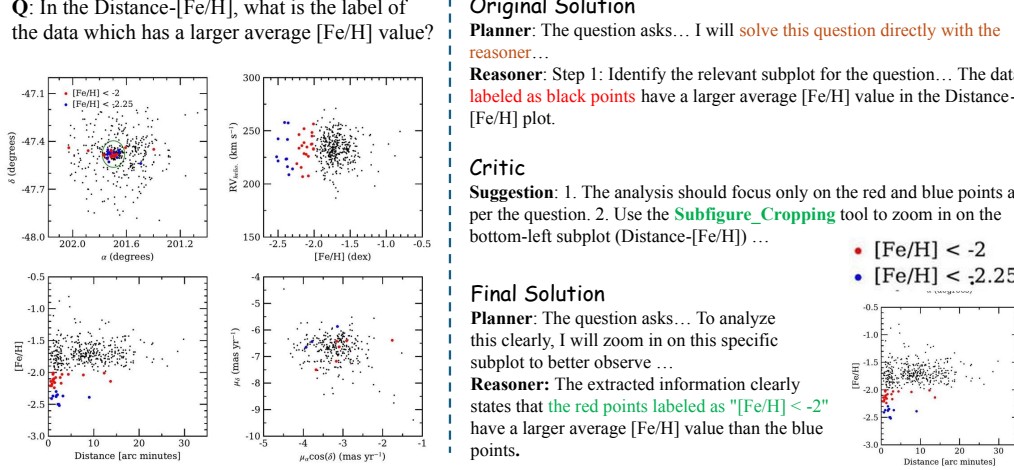

**Q**: In the Distance-[Fe/H], what is the label of the data which has a larger average [Fe/H] value?

**Original Solution**
**Planner**: The question asks… I will solve this question directly with the reasoner…
**Reasoner**: Step 1: Identify the relevant subplot for the question… The data labeled as black points have a larger average [Fe/H] value in the Distance-[Fe/H] plot.

**Critic**
**Suggestion**: 1. The analysis should focus only on the red and blue points as per the question. 2. Use the **Subfigure_Cropping** tool to zoom in on the bottom-left subplot (Distance-[Fe/H]) …

**Final Solution**
**Planner**: The question asks… To analyze this clearly, I will zoom in on this specific subplot to better observe …
**Reasoner:** The extracted information clearly states that the red points labeled as "[Fe/H] < -2" have a larger average [Fe/H] value than the blue points.

Figure 8: Illustration of self-correction.

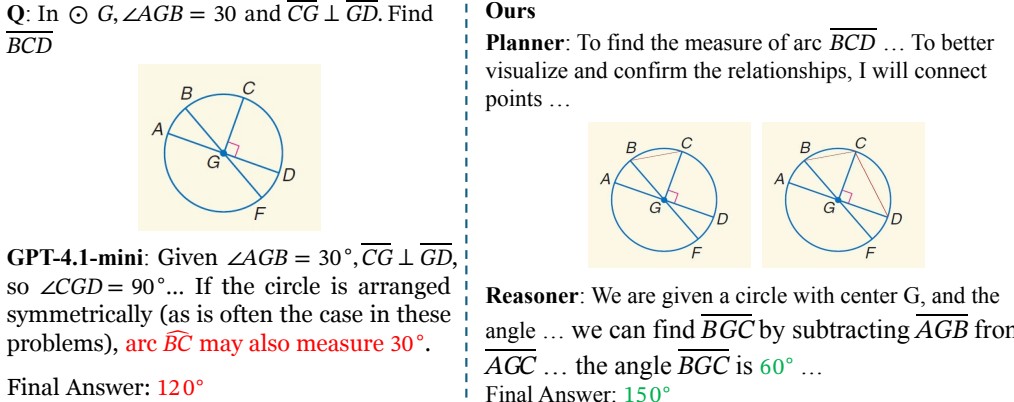

**Q**: In ⊙ $G$, $\angle AGB = 30$ and $\overline{CG} \perp \overline{GD}$. Find $\overline{BCD}$

**GPT-4.1-mini**: Given $\angle AGB = 30°$, $\overline{CG} \perp \overline{GD}$, so $\angle CGD = 90°$… If the circle is arranged symmetrically (as is often the case in these problems), arc $\widehat{BC}$ may also measure 30°.

Final Answer: 120°

**Ours**
**Planner**: To find the measure of arc $\overline{BCD}$ … To better visualize and confirm the relationships, I will connect points …

**Reasoner**: We are given a circle with center $G$, and the angle … we can find $\overline{BGC}$ by subtracting $\overline{AGB}$ from $\overline{AGC}$ … the angle $\overline{BGC}$ is 60° …
Final Answer: 150°

Figure 9: Illustration of Geometric Reasoning.

# D    ADDITIONAL RESULTS

## D.1    SELF-CORRECTION EXAMPLES

We present an illustrative case to demonstrate the system's self-correction capabilities. In Fig. 8, the critic identified that the initial reasoning omitted the *Subfigure_Cropping* tool. Based on this feedback, the planner refined its process, leading to the correct answer.

## D.2    GEOMETRIC REASONING EXAMPLES

As shown in Fig. 9, we present a representative case to illustrate our system's ability in geometric reasoning. Our visual tools further enable the model to faithfully capture point–line relationships in geometric diagrams, thereby enhancing the accuracy and robustness of its reasoning performance.

## D.3    ADDITIONAL GROUNDING RESULTS

To demonstrate the effectiveness of our finetuned grounding model, we compare its performance with the base model on a test set of 500 samples. This test set was annotated using the same pipeline as our synthetic training data to ensure consistency. As summarized in the main text and detailed in Table 6, our finetuned model achieves a dramatic improvement in grounding accuracy. The overall

Table 6: **Grounding accuracy on structured chart elements.** IoU columns evaluate bounding-box overlap for subplots, legend regions, and textual labels (titles and axis labels). PCK@0.01 measures point localization accuracy for axis tick marks using a threshold of $0.01 \times \max(\text{height}, \text{width})$.

| Model | Subplot regions (IoU) | Legend regions (IoU) | Text labels (IoU) | Axis ticks (PCK@0.01) | Overall |
|---|---|---|---|---|---|
| Qwen2.5-VL-3B | 0.27 | 0.04 | 0.05 | 0.04 | 0.10 |
| Qwen2.5-VL-7B | 0.52 | 0.17 | 0.21 | 0.15 | 0.26 |
| GPT-4.1-mini | 0.78 | 0.18 | 0.11 | 0.17 | 0.31 |
| Ours (3B) | 0.99 | 0.89 | 0.90 | 0.93 | 0.93 |

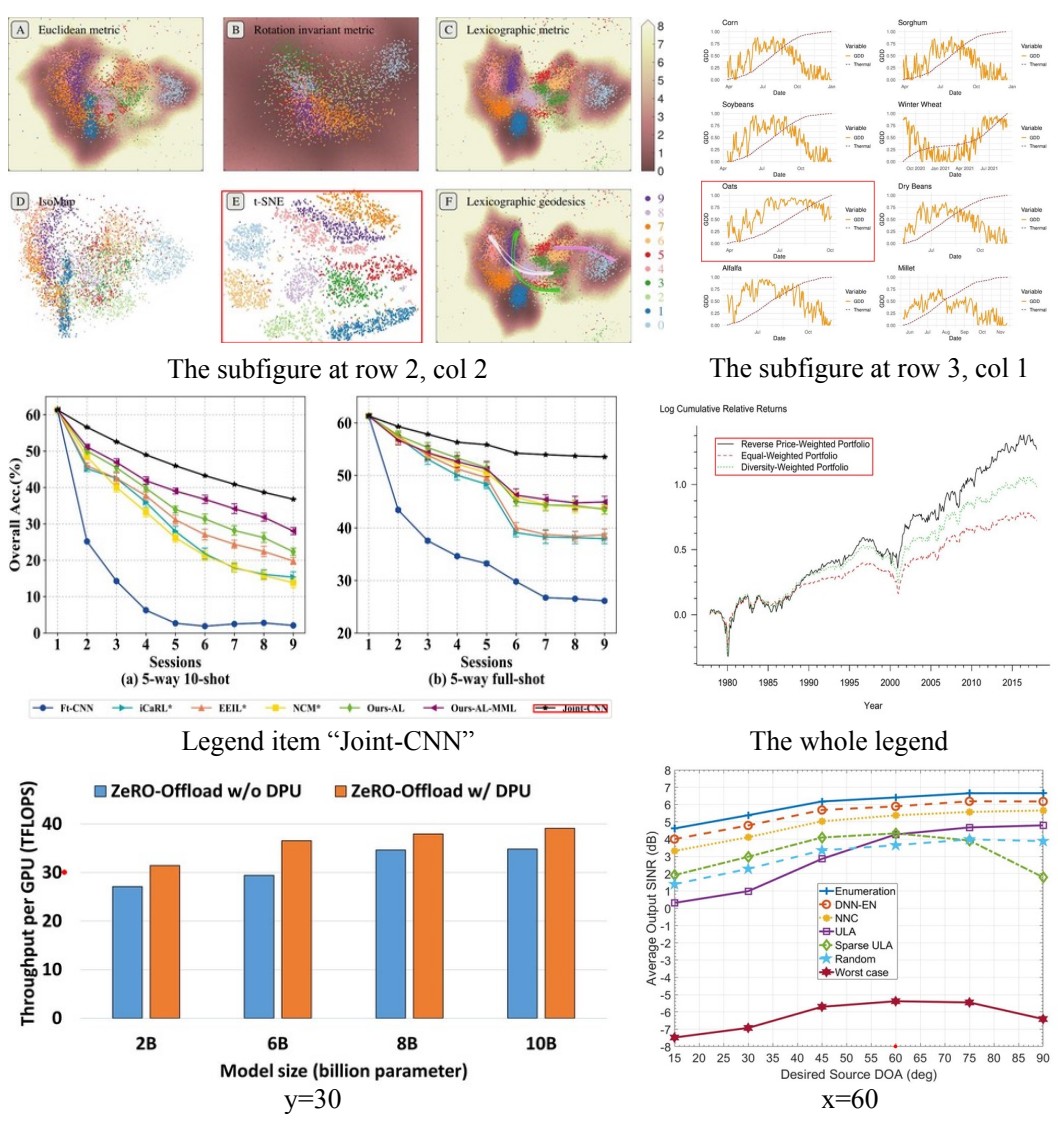

The subfigure at row 2, col 2          The subfigure at row 3, col 1

Legend item "Joint-CNN"          The whole legend

y=30          x=60

Figure 10: Additional grounding examples.

accuracy for our model is 0.93, a substantial leap from the base model's 0.26. Additional visual examples can be found in Fig. 10. This precise localization of chart components is the foundation upon which our system's downstream chart analysis capabilities are built.

# E    COMPREHENSIVE ANALYSIS OF COMPUTATIONAL COST AND EFFICIENCY

In this section, we provide a detailed analysis of PixelCraft's computational overhead. We first compare our method with existing visual tool and multi-agent baselines in terms of latency and API calls. Subsequently, we compare PixelCraft against Self-Refine strategies (Test-Time Scaling) to demonstrate that our performance gains stem from the proposed architecture rather than merely increased computational budget.

## E.1    COMPARISON WITH VISUAL TOOL AND MULTI-AGENT BASELINES

There is generally a trade-off where agentic frameworks achieve better performance by incurring higher computational costs due to iterative planning and tool execution. To demonstrate PixelCraft's relative efficiency, we compare it with both Visual Tool Baselines (Refocus and Visual CoT) and Multi-Agent Baselines (Debate and Reconcile) on the CharXiv benchmark.

We report the average inference latency (seconds per query), the average number of API calls, and the corresponding accuracy in Table 7. Note that "Visual CoT" is defined under the same setting as in Section 5.3 and refers to a linear agent baseline equipped with visual tools and indiscriminately includes all historical images in the context.

Table 7: Computational cost analysis comparing PixelCraft with visual tool and multi-agent baselines on CharXiv. Latency is measured in seconds per query.

| Method | Latency (s) | API Calls | Accuracy (%) |
|---|---|---|---|
| CoT | 3.75 | 1.0 | 63.8 |
| Refocus | 14.31 | 2.0 | 60.7 |
| Visual CoT | 19.31 | 4.4 | 65.0 |
| Debate | 27.65 | 7.0 | 62.4 |
| Reconcile | 25.90 | 6.0 | 63.5 |
| **Ours** | 16.45 | 5.7 | **68.1** |

**Scalability vs. Multi-Agent Systems.**    As shown in Table 7, PixelCraft demonstrates superior efficiency compared to existing multi-agent frameworks, reducing latency by approximately 36% to 40% relative to Reconcile and Debate. This efficiency stems from PixelCraft's *adaptive workflow*. Unlike Debate or Reconcile, which typically enforce a fixed workflow (e.g., fixed debate rounds) regardless of query complexity, our Planner can terminate the process early for simpler queries. This adaptability avoids the computational overhead inherent in rigid multi-agent interactions.

**Comparison with Linear Agent (Visual CoT).**    Crucially, PixelCraft achieves lower latency than the Visual CoT baseline (16.45s compared to 19.31s). This validates the computational advantage of our **Image Memory** design. Unlike the linear Visual CoT approach, which stacks the entire history of generated images into the context window (increasing token count and processing time), our planner selectively recalls only the necessary visual clues. This ensures a more compact context and faster inference without sacrificing reasoning depth.

**Trade-off Analysis.**    While PixelCraft incurs a slight increase in latency compared to the simpler Refocus method (+2.14s), it yields a substantial performance gain (+7.4% on CharXiv). Regarding API calls, our design introduces a dispatcher, image memory, and critics to reduce the planner's decision burden. Although this yields more API calls than single-turn methods, the context per call is much shorter, resulting in superior latency efficiency compared to baseline systems.

## E.2    COMPARISON WITH SELF-REFINE BASELINES (TEST-TIME SCALING)

To rigorously validate that our performance gains are driven by our specific multi-agent design rather than simply increased test-time compute, we compare PixelCraft against a Self-Refine baseline.

**Experimental Setup.** We implemented a Self-Refine baseline following the workflow proposed in *Critic-V* (Zhang et al., 2025a). The workflow involves a Reasoner generating an answer and a Critic providing feedback to refine it. To ensure a fair, cost-aligned comparison, we configured the Self-Refine baseline to run for 2 rounds (approx. 5 calls) and 3 rounds (approx. 7 calls), which brackets the average API calls of PixelCraft (5.7).

Table 8: Comparison with Self-Refine baselines (Test-Time Scaling) on CharXiv and ChartQAPro.

| Method | Cost (Approx. Calls) | CharXiv | ChartQAPro |
|---|---|---|---|
| CoT | 1 | 63.8 | 62.21 |
| Self-Refine (1 round) | 3 | 61.4 | 63.14 |
| Self-Refine (2 rounds) | 5 (Similar to Ours) | 62.0 | 62.47 |
| Self-Refine (3 rounds) | 7 | 61.0 | 61.60 |
| **Ours** | $\sim$5.7 | **68.1** | **65.56** |

**Analysis.** As shown in Table 8, simply applying self-refinement strategies did not yield consistent performance gains and even caused degradation on CharXiv (63.8 $\rightarrow$ 62.0). This observation aligns with recent findings in *Sherlock* (Ding & Zhang, 2025), which reported that VLM self-refinement often degrades performance on reasoning benchmarks due to hallucinated critiques.

The significant gap between PixelCraft (68.1%) and Self-Refine (62.0%) under the same compute budget demonstrates that **access to high-fidelity tools and a flexible reasoning paradigm** rather than just more reasoning turns—is the key to overcoming perceptual bottlenecks in structured image reasoning.

## F    ADDITIONAL EVALUATION ON REAL-WORLD INFOGRAPHICVQA

To evaluate the generalization capability of PixelCraft on open-domain, real-world structured images beyond standard chart benchmarks, we conducted additional experiments on the **InfographicVQA** (Mathew et al., 2022) benchmark. We reuse the same configuration as in the chart settings, without introducing any additional tools.

### F.1    EXPERIMENTAL SETUP

**Dataset Filtering.** InfographicVQA contains a mix of document images, some of which require only optical character recognition (OCR) on dense text, while others involve complex visual reasoning over infographics, charts, and tables. Since our work focuses on *structured image reasoning*, we curated a relevant subset from the InfographicVQA validation set.

We employed GPT-5-mini as a filter to identify questions specifically requiring reasoning over charts, tables, or graphic layouts, **excluding queries that rely solely on reading text blocks**. This process yielded a challenging subset of **947 samples**. The prompt used for this filtering process is provided below to ensure reproducibility:

---

**Filtering Prompt for InfographicVQA Subset**

- **System:** You are a helpful AI assistant. Given **an image** (which may contain charts, tables, diagrams, screenshots, or plain text) and **a question about that image**, decide whether answering the question truly **requires understanding chart/table data**, or if it can be answered **only by reading/extracting text**. Your goal is to select a subset of **chart/table-related** examples.
  Follow these rules:

  1. **Chart/Table-related (select = true)** Set "select": true **only if** the question mainly requires understanding the **data structure or quantitative information** in a chart or table. Typical signs: 1) Interpreting axes, scales, bars, lines, points, or cells. 2) Comparing values (higher/lower, increase/decrease, trends, patterns). 3) Aggregating or

---

combining numbers (totals, differences, ratios, averages, etc.). 4) Understanding the layout/structure of a table (row/column relationships).

2. **Text-only (select = false)** Set "select": false if the question can be answered by simply **reading text** in the image, without using any chart/table structure. Examples: 1) Asking about titles, captions, legend labels, axis labels. 2) Asking for a single printed number that requires no comparison or trend analysis. 3) Asking about UI elements (buttons, menus, logos). 4) Anything solvable via OCR-style text extraction only.

3. **Special cases** 1) If the image does not contain a chart or table → '"select": false'. 2) If the question is unrelated to the chart/table or the image content → "select": false. 3) If uncertain, be conservative → "select": false.

- **User:** (`example.png`) Please decide whether the above image should be selected given the question: {question}

## F.2 QUANTITATIVE RESULTS

We compared PixelCraft against standard Chain-of-Thought (CoT) and multi-agent baselines (Debate, Reconcile) on this subset. To handle the diverse visual styles and ensure better generalization on these real-world images, we employed a Qwen2.5-VL-7B grounding model for this experiment, which was fine-tuned on the exact same training data as the 3B model used in our main evaluation. As shown in Table 9, PixelCraft achieves consistent performance gains across different backbones.

Table 9: Performance comparison on the structured reasoning subset of InfographicVQA. PixelCraft consistently outperforms baselines, demonstrating strong generalization to real-world infographics.

| Method | GPT-4.1-mini | GPT-4o |
|---|---|---|
| CoT | 77.3 | 69.9 |
| Debate | 74.0 | 67.6 |
| Reconcile | 73.7 | 67.3 |
| **Ours** | **79.8** | **71.6** |

Unlike structured charts, real-world infographics often contain noisy layouts and diverse visual styles. The superior performance of PixelCraft suggests that our grounding-based tool agents effectively handle these complexities, reducing perceptual errors where standard VLM reasoning fails.

## F.3 QUALITATIVE ANALYSIS

To better understand the performance gains shown in Table 9, we visualize representative reasoning trajectories in Figure 11. These examples highlight how PixelCraft overcomes the perceptual bottlenecks inherent in standard Chain-of-Thought (CoT).

## G LIMITATIONS AND FUTURE WORKS

While PixelCraft significantly improves performance across several challenging benchmarks, we identify two key limitations that highlight promising directions for future work.

One limitation stems from the current inability of MLLMs to autonomously generate reliable, high-fidelity visual tools. Our initial explorations found that purely LLM-generated tools were often ineffective, suffering from incorrect code execution for visual editing, which requires slight manual validation and refinement (see details in Appendix B). Future work on fully automated tool generation and verification is thus critical to improve adaptability and reduce this curation overhead.

Additionally, PixelCraft requires strong backbone MLLMs for task decomposition and tool orchestration. Weaker ones may constrain the framework's effectiveness with incorrect planning and tool calling. Future work could investigate mitigation strategies, such as developing more robust agent communication protocols or lightweight, specialized models for planning and criticism.

**Q**: Which is the major city in **Province 5**?

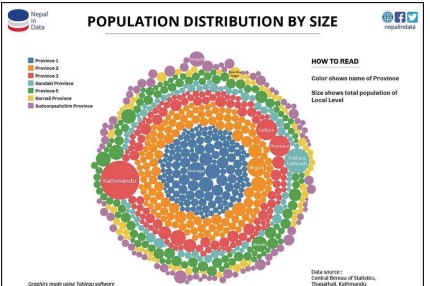

**CoT**

Step 1: Identify the color … Answer: The major city in Province 5 is Pokhara Lekhnath.

**Ours**

To find the major city in Province 5, I need to focus on … Masking Data with Legend … Final Answer: The major city in Province 5 is Ghorahi.

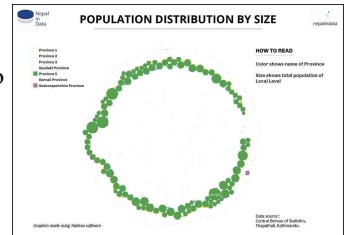

**Q**: How many companies constitute the Disney Theatrical group?

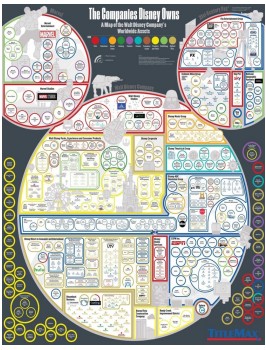

**CoT**

The Disney Theatrical Group consists of the following companies … there are 4 companies/entities.

**Ours**

The question asks for the number of companies … Subfigure Cropping … Final Answer: The Disney Theatrical Group consists of 8 companies.

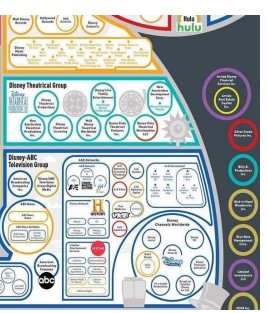

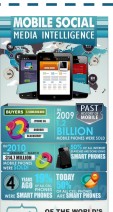
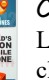

**Q**: By what colour is IPHONE 3G in pie chart represented-yellow, green or purple?

**CoT**

Let's analyze the pie chart …Therefore, the IPHONE 3G in the pie chart is represented by the yellow color.

**Ours**

The question asks about the color … Subfigure Cropping …

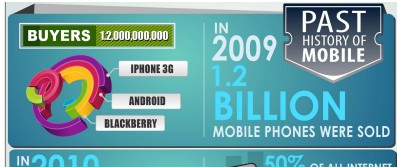

Final Answer: IPHONE 3G is represented by the purple color in the pie chart.

Figure 11: **Qualitative Examples on InfographicVQA.** PixelCraft effectively selects and utilizes tools (e.g., Subfigure Cropping, Masking Data with Legend) to answer questions requiring detailed inspection of complex infographics, even when visual inputs are less structured.

Progress in these complementary directions will be critical for evolving PixelCraft into a more general and adaptable framework for structured image reasoning.

