# OpenReview forum: "PixelCraft: A Multi-Agent system for High-Fidelity Visual Reasoning on Structured Images"
_ICLR.cc/2026/Conference — ICLR 2026 Poster_

### Official Review · Reviewer_hJXi · 2025-10-27

**Soundness:** 3
**Presentation:** 3
**Contribution:** 2
**Rating:** 4
**Confidence:** 5

**Summary:**

This paper presents PixelCraft, a novel multi-agent system designed to address significant failings in MLLM-based visual reasoning for structured images like charts and geometric diagrams. The authors identify two primary weaknesses in existing methods: low-fidelity image processing, which leads to perceptual errors, and rigid, linear reasoning patterns (like visual CoT) that cannot correct mistakes. PixelCraft's core contribution is a dual-pronged solution. First, it achieves high-fidelity processing by synergizing a compact MLLM, fine-tuned on a new synthetic corpus for precise pixel-level grounding, with classical CV algorithms that act as tool agents. Second, it enables
flexible, non-linear reasoning through a dynamic workflow involving a planner, reasoner, and critics. A key innovation is the "image memory," which allows the planner to adaptively revisit, recall, and branch from earlier visual steps, facilitating backtracking and exploration of alternative reasoning paths. The authors demonstrate through extensive experiments on challenging chart and geometry benchmarks that their system significantly improves reasoning accuracy for advanced MLLMs.

**Strengths:**

1. Well-motivated and significant problem. The paper addresses the challenging problem of visual reasoning on structured images, a well-known weak point for current MLLMs. This task is difficult because it requires precise perception of details (like axis values or geometric points) and multi-step logical deduction, where small perceptual errors can cascade into completely wrong answers. The paper clearly identifies these failings as a critical and practical area for improvement.

2. Comprehensive experimental setup. The authors validate their system on a strong suite of recent and difficult benchmarks, including CharXiv, ChartQAPro, and the auxiliary-line subset of Geometry3K. Testing against these diverse datasets demonstrates the system's robustness and generalizability. Furthermore, the comparison is not limited to simple baselines; it includes advanced agentic methods like Debate and Reconcile, which makes the consistent and significant performance gains of PixelCraft more convincing.

3. High-quality presentation and readability. The paper is very easy to read and well-organized, making the complex multi-agent architecture and reasoning process accessible to the reader. The figures are a standout component for presentation; Figure 1 provides an excellent, intuitive comparison of why standard CoT and Visual CoT fail while PixelCraft succeeds, and Figure 2 clearly illustrates the entire agentic workflow, from selection to discussion and correction.

**Weaknesses:**

1. Significant efficiency and latency issues. The multi-agent workflow, which involves a dispatcher, planner, reasoner, and multiple critics, inherently requires numerous sequential LLM inference calls for a single query. This leads to a substantial increase in latency and computational cost compared to simpler CoT methods, as confirmed by the paper's own analysis (e.g., 16.45s vs. 3.75s on CharXiv), which may limit its practical applicability.

2. Limited methodological novelty in the agentic framework. While the application to structured images is effective, the core agentic workflow (combining a planner, tool-use, and a reasoner) is a widely adapted paradigm in recent LLM research. The novelty of the agent framework itself is somewhat incremental, with the main contributions lying in the specialized toolset and its application rather than a fundamentally new agentic reasoning process.

3. Constrained generalizability and flexibility due to a fixed toolset. The system's success is heavily reliant on its manually-curated set of visual tools, which were specifically designed for charts and geometric diagrams. This presents a generalization bottleneck; the system would likely fail in scenarios with novel image types or tasks requiring tools not in its predefined set. This lack of flexibility to adapt or generate new tools on the fly is a key limitation, which the authors also acknowledge.

**Questions:**

N/A

---

> ### Author Response · Authors · 2025-11-21
> **Response to Reviewer hJXi (Part 1/3)**
>
> Dear Reviewer hJXi,
>
> We thank the reviewer for the thoughtful evaluation, and for highlighting the significance of structured-image reasoning, the strength of our experimental study, and the clarity of our multi-agent presentation. We also appreciate the constructive comments on efficiency, agentic novelty, and toolset generalization, which we address in detail below.
>
> > **Response to W1: Significant efficiency and latency issues**
>
> Please refer to the general response for further details, where we report latency and average number of API calls comparison with baselines, which shows that PixelCraft is faster than most baseline systems. In summary, PixelCraft optimizes the balance between cost and performance, achieving substantial accuracy gains over prior systems while maintaining competitive efficiency among agentic frameworks. This efficiency comes from our dedicated agent design, which reduces per-step context and avoids rigid multi-agent rounds while improving reasoning quality.

---

> ### Author Response · Authors · 2025-11-21
> **Response to Reviewer hJXi (Part 2/3)**
>
> > **Response to W2: Limited methodological novelty in the agentic framework.**
>
> We acknowledge that planner–tool–reasoner style agent frameworks have begun to appear in recent LLM work. However, we respectfully argue that adoption of a paradigm does not equate to a lack of novelty. Our contribution is not the template itself, but the restructuring of the reasoning process to address the unique challenges of structured images (distributed evidence, lossy tool outputs) that standard linear agents fail to handle.
>
> We clarify our novelty regarding the Agentic Reasoning Process below:
>
> **1. Evidence that the gains are NOT just from the tools.**
>
> The reviewer suggests improvements might come primarily from the toolset. However, our ablation results suggest that the agentic framework itself contributes substantially. As shown in Table 3 (Sec. 5.3), even with exactly the same toolset, our framework outperforms the prior Visual CoT agent by +3.10 on CharXiv and +4.52 on ChartQAPro. This confirms that the gains are driven by our novel reasoning architecture—specifically how the agent coordinates planning and verification—rather than the tools alone.
>
> **2. Methodological Novelty.**
>
> Existing methods in visual reasoning, whether relying on single-agent Chain-of-Thought or multi-agent architectures, are fundamentally constrained by **rigid visual reasoning process**. This linear, rigid execution or workflow often fail to support the queries that require flexible, tree-like visual reasoning process. Our framework introduces a paradigm shift through two interconnected contributions: non-linear visual reasoning via selective recall, and a dynamic agentic workflow that realizes this non-linearity in practice.
>
> - **From Linear Execution to Flexible Non-Linear Visual Reasoning.**
> Unlike prior works (e.g., Visual SKETCHPAD (NeurIPS 2024) [1], Refocus (ICML 2025) [2]) that
> follow a rigid visual sequence, our reasoning process is explicitly non-linear and supports dynamic branching and backtracking.
>
>     **How we achieve this:** We introduce a **Compact Visual Memory** mechanism. Instead of feeding all historical images into the context (which confuses the LLM), the planner selectively recalls intermediate visual clues to revisit earlier steps. This transforms the process from a static script into a dynamic recall for visual reasoning.
>
> - **From Rigid Multi-Agent Workflow to Dynamic Agentic Workflow**.
> Building on the above shift in visual reasoning, we also rethink the system-level control flow, enabling the agent to branch, backtrack, and re-plan on demand. While standard multi-agent paradigms generally enforce a fixed communication and execution workflow (e.g., MDAgents (NeurIPS 2024 oral) [3], Agent Reviewers (ICML 2025) [4]), ours is dynamic.
>
>     **How we achieve this**: We decompose the workflow into three atomic actions: (i) high-level planning, (ii) tool agent selection and execution, and (iii) visual reasoning. The planner dynamically chooses the next action (calling visual tools or invoking the reasoner), and these components are adaptively scheduled based on query complexity. With this design, complex queries trigger a deeper chain of image processing steps, while simple queries allow for immediate answer generation without unnecessary computation.
>
> **3. How novelty is recognized in agentic literature.**
>
> We would like to emphasize that, in the agent literature, novelty does not only come from proposing a fundamentally new agentic reasoning process; it also lies in how a generic template is instantiated and adapted to a challenging domain, which is itself regarded as a substantial contribution. Recent works such as MDAgents (NeurIPS 2024 oral), FinCon (NeurIPS 2024) [5], and Agent Reviewers (ICML 2025) are widely recognized for their domain-specific agent designs built on shared high-level patterns. Our work is in this line: we contribute a structured-image–specific agent architecture whose design choices are both principled and empirically validated.
>
> Beyond the agent design, our toolset itself is novel. Please see our response to the next point for details on the toolset’s generalizability.
>
> ### References
> [1] Hu, Yushi, et. al. Visual sketchpad: Sketching as a visual chain of thought for multimodal language models. NeurIPS 2024.
>
> [2] Fu, Xingyu, et. al. Refocus: Visual editing as a chain of thought for structured image understanding. ICML 2025.
>
> [3] Kim, Yubin, et. al. Mdagents: An adaptive collaboration of llms for medical decision-making. NeurIPS 2024.
>
> [4] Lu, Kai, et. al. Agent Reviewers: Domain-specific Multimodal Agents with Shared Memory for Paper Review. ICML 2025.
>
> [5] Yu, Yangyang, et. al. Fincon: A synthesized llm multi-agent system with conceptual verbal reinforcement for enhanced financial decision making. NeurIPS 2024.
>
> [6] Hong, Jack, et. al. DeepEyesV2: Toward Agentic Multimodal Model. arXiv 2025.

---

> ### Author Response · Authors · 2025-11-21
> **Response to Reviewer hJXi (Part 3/3)**
>
> > **Response to W3: Constrained generalizability and flexibility due to a fixed toolset**
>
> We thank the reviewer for raising this important point regarding generalizability and flexibility. We agree that our current toolset is instantiated for charts and geometric figures, and that building high-fidelity visual tools for structured-image reasoning remains a fundamental open challenge. Existing systems at top venues either rely on underlying ground-truth codes (e.g., Visual SKETCHPAD) or on fully hand-crafted, low-fidelity operations (e.g., Refocus), while very recent work such as DeepEyesV2 (Nov. 2025) [6] auto-generates code at inference time for only simple primitives like cropping and region marking. In contrast, our high-fidelity tools are not manually crafted from scratch, but rather derived through a **semi-automated, data-driven generation pipeline** described in **Appendix B**.
>
> - **Scalable Tool Generation Pipeline:** As detailed in **Appendix B**, we employ a “Generation–Clustering–Refinement” pipeline. Given a dataset, an LLM first proposes 468 candidate tools. It then clusters these candidates and selects the most common tools (e.g., cropping, drawing auxiliary lines). Finally, it abstracts the selected tools into more general forms to improve robustness and generalization, with final human refinement for some complex tools.
> - **Reduced Human Intervention:** While we acknowledge that a final round of expert verification was performed to ensure precision for some complex tools, this data-driven pipeline **significantly reduces the manual effort and reliance on expert knowledge** required to design tool sets.
> - **Generalization to New Domains:** First, our toolset shows promising generalization to new domains. On InfographicsVQA, a challenging real-world document-understanding benchmark, we reuse exactly the same tool configuration as in the chart setting, without introducing any additional tools. As shown in the below table, PixelCraft still outperforms CoT and other agentic baselines. Please refer to Appendix F in the revised manuscript for detailed results and qualitative case studies.
>
>     | Method            | InfoVQA (GPT-4.1-mini) | InfoVQA (GPT-4o) |
>     |-------------------|-------------------------|-------------------|
>     | CoT               | 77.3                    | 69.9              |
>     | Debate            | 74.0                    | 67.6              |
>     | Reconcile         | 74.7                    | 67.3              |
>     | PixelCraft (Ours) | **79.8**                | **71.6**          |
>
> Moreover, extending PixelCraft to a completely new domain (e.g., **medical imaging**) does not require from-scratch manual tool engineering; one can instead run our semi-automated pipeline to automatically discover and then semi-automatically refine domain-specific tools (e.g., `segment_lesion`, `measure_diameter`, or `locate_organ_region`) suitable for that task.

---

> ### Author Response · Authors · 2025-11-27
>
> Dear Reviewer hJXi:
>
> Thank you again for your thoughtful feedback and for taking the time to review our work.
>
> We hope that our responses have addressed your concerns. As the rebuttal deadline is approaching, we are writing to kindly check whether our replies have resolved your questions or if any points would benefit from further discussion.
>
> Please feel free to let us know if any further questions arise.

---

### Official Review · Reviewer_tt9L · 2025-10-31

**Soundness:** 3
**Presentation:** 4
**Contribution:** 4
**Rating:** 8
**Confidence:** 3

**Summary:**

This paper proposes PixelCraft, a multi-agent system to perform complex reasoning over charts and geometric figures. The multi-agent system features a tool selector, a tool planner, visual critic, plan critic and multiple tool agents. Experiments on chart reasoning benchmarks like ChartXiv and ChartQAPro, and geometry reasoning benchmarks shows substantial improvement over other framework or vaniila visual CoT.

**Strengths:**

+ The paper proposes a novel multi-agent system to improve visual reasoning performance on structured image.
+ Achieves high-fidelity image process with a fine-tuned grounding model
+ The paper is well written and easy to follow.

**Weaknesses:**

+ My major concern is the cost of the proposed multi-agent system compared with vanilla visual CoT. As shown in Appendix D.4, the response time of the multi-agent system is approximately three or four times longer than baseline methods. Therefore, it would be helpful to reveal the cost (or the number of generated tokens) of PixelCraft in comparision with baseline method.

+ Moreover, it is not clear if the compared baseline methods also use a critic to iteratively refine and improve the final result. If not, I would suggest equipping the baseline methods with a critic to do test-time scaling, and draw a comparison when the respond time/cost is similar.

**Questions:**

See the weaknesses above.

---

> ### Author Response · Authors · 2025-11-21
> **Response to Reviewer tt9L**
>
> Dear Reviewer tt9L,
>
> We thank the reviewer for the positive assessment and for highlighting the novelty of our multi-agent system and its high-fidelity processing capabilities. We appreciate the constructive suggestions regarding cost analysis and test-time scaling comparisons, which we address below.
>
> > **Response to W1: Cost Analysis**
>
> Please refer to the general response for further details, where we report latency and average number of API calls comparison with Visual Tool Baselines and Multi-Agent Baselines. In summary, PixelCraft optimizes the balance between cost and performance, achieving substantial accuracy gains over prior systems while maintaining competitive efficiency among agentic frameworks.
>
> > **Response to W2: Comparison with Self-Refine Baselines (Test-Time Scaling)**
>
> We appreciate this insightful suggestion. To rigorously validate our architecture, we compare PixelCraft against a baseline that also utilizes iterative refinement (test-time scaling) to ensure the gains are not achieved by increased compute but by dedicated critic design in our framework.
>
> **1. Experimental Setup (Cost-Aligned Comparison):**
> To ensure a fair comparison at a **similar computational budget**, we implemented a **Self-Refine Baseline** following the workflow proposed in **Critic-V** [1].
>
> - **Method:** This baseline employs a `Reasoner` to generate an initial answer, followed by a `Critic` that provides feedback based on the image and question, prompting the `Reasoner` to refine its answer.
> - **Cost Alignment:** PixelCraft uses an average of **5.7 API calls** per query. To match this, we configured the Self-Refine baseline to run for **2/3 refinement rounds** (totaling approx. 5/7 API calls per query).
>
> **2. Results:**
> We evaluated this baseline on CharXiv and ChartQAPro. The results are compared below:
>
> | Method               | Cost (Approx. Calls) | CharXiv | ChartQAPro |
> |----------------------|-----------------------|---------|------------|
> | CoT (Baseline)       | 1                     | 63.8    | 62.21      |
> | Self-Refine (1 round) | 3                     | 61.4    | 63.14      |
> | Self-Refine (2 rounds) | 5 (Similar to Ours)  | 62.0    | 62.47      |
> | Self-Refine (3 rounds) | 7                     | 61.0    | 61.60      |
> | PixelCraft (Ours)    | ~5.7                  | **68.1**    | **65.56**      |
>
> **3. Analysis and Discussion:**
>
> - **Ineffectiveness of Naive Self-Refinement:** As shown in the table, simply applying self-refinement strategies (like those in Critic-V) to these benchmarks **did not yield performance gains**, and even caused a performance drop on CharXiv (63.8 $\rightarrow$ 62.0). This observation aligns with recent findings in [2], which reported that VLM self-refinement often degrades performance on reasoning benchmarks.
> - **Value of PixelCraft:** This comparison strongly highlights the necessity of our approach. The significant gap between PixelCraft (68.1) and Self-Refine (62.0) under the same compute budget demonstrates the advantage of PixelCraft rather than just more reasoning turns.
>
> This comparison has been added to Appendix E.2.
>
> ### References
> [1] Zhang, Di, et. al. Critic-v: Vlm critics help catch vlm errors in multimodal reasoning. CVPR 2025.
>
> [2] Ding, Yi, et. al. Sherlock: Self-Correcting Reasoning in Vision-Language Models. arXiv 2025.

---

> ### Author Response · Authors · 2025-11-27
>
> Dear Reviewer tt9L:
>
> Thank you again for your thoughtful feedback and for taking the time to review our work.
>
> We hope that our responses have addressed your concerns. As the rebuttal deadline is approaching, we are writing to kindly check whether our replies have resolved your questions or if any points would benefit from further discussion.
>
> Please feel free to let us know if any further questions arise.

---

### Official Review · Reviewer_yBc8 · 2025-11-01

**Soundness:** 3
**Presentation:** 3
**Contribution:** 3
**Rating:** 6
**Confidence:** 3

**Summary:**

The paper presents a new framework designed to improve how multimodal models reason about structured visual content such as charts, diagrams, and geometry problems. It introduces a multi-agent architecture where specialized agents, like a planner, reasoner, and visual tool users, collaborate to analyze images, exchange feedback, and iteratively refine their reasoning. A key feature is a pixel-level grounding module, which allows the system to interact precisely with image regions through actions like zooming, masking, and highlighting. The framework also incorporates an image memory and planning mechanism that enables non-linear reasoning and self-correction. Experiments show that PixelCraft achieves strong performance across visual reasoning benchmarks, outperforming previous multi-agent and single-model approaches.

**Strengths:**

1. Originality: This paper integrates multi-agent collaboration with fine-grained visual grounding for structured image reasoning. While the idea of multi-agent reasoning is not entirely new, combining it with pixel-level visual tools and planner-managed image memory adds a modest but meaningful improvement in coordination and interpretability.

2. Quality: The work is technically sound and experimentally thorough. The system design is coherent, and the experiments cover diverse benchmarks with reasonable comparisons and ablations.

3. Clarity: The paper is generally well organized, with clear descriptions of each agent’s role and effective figures that illustrate the workflow.

4. Significance: The contributions are incremental but relevant. The framework addresses practical challenges in visual reasoning and could be adapted to other structured domains, offering a small yet useful step forward for multimodal systems.

**Weaknesses:**

1. Unclear Cost and Scalability

The multi-agent architecture (planner, critics, tool agents, etc.) implies high computational and communication overhead. The paper doesn’t report inference latency, average agent calls, or cost per sample — critical factors for assessing practical deployability. Since they use a strong vision-language backbone (Qwen2.5-VL), much of the pipeline’s gain might come at the cost of efficiency. Report computation and cost metrics, average number of reasoning turns, total API calls, and latency per image — to give a fair sense of scalability.

2. Overreliance on Synthetic or Narrow Benchmarks

Most experiments use structured-image datasets (e.g., ChartQA, Geometry3K), which are quite specialized. This limits generalization to open-domain or real-world structured reasoning tasks like document or diagram understanding. It’s not shown whether PixelCraft scales to noisy or natural images where structure is less explicit. I suggest adding a small evaluation on broader multimodal datasets (e.g., DocVQA, InfographicsVQA) or including a discussion on how the system might handle unstructured visuals.

**Questions:**

I am confused about the Figure. 4. Why is the usage frequency of Legend Masking and Adding Auxiliary very low, but the performance gain is giant? Most of the time, these tools will not be called.

---

> ### Author Response · Authors · 2025-11-21
> **Response to Reviewer yBc8**
>
> Dear Reviewer yBc8,
>
> We thank the reviewer for the thoughtful assessment and for recognizing the originality of our high-fidelity visual grounding approach as well as the coherence of our system design. We appreciate the constructive feedback regarding scalability and generalization. We address these points and the clarification regarding Figure 4 below.
>
> > **Response to W1: Unclear Cost and Scalability**
>
> **A1**: Please refer to the general response for further details, where we report latency and average number of API calls comparison with Visual Tool Baselines and Multi-Agent Baselines. Additionally, we would like to point out that our grounding model is a compact fine-tuned Qwen2.5-VL-3B, which incurs only a small latency overhead compared with API calls using larger advanced LLMs. In summary, PixelCraft optimizes the balance between cost and performance, achieving substantial accuracy gains over prior systems while maintaining competitive efficiency among agentic frameworks.
>
> > **Response to W2: Overreliance on Synthetic or Narrow Benchmarks**
>
> We agree that evaluating on broader, open-domain datasets is crucial to assess robustness. To demonstrate PixelCraft's generalization capability beyond specialized charts and geometry, we evaluate our method on **InfographicsVQA**, a challenging real-world document understanding benchmark.
>
> - **Experimental Setup:** To focus the evaluation on visual reasoning (the core contribution of our paper) rather than pure text extraction, we filtered for chart and table-related questions (947 samples) from InfographicsVQA. To ensure transparency and reproducibility, the specific prompt used for this filtering process is provided in the revised Appendix F.1.  To demonstrate that our approach generalizes to noisy, real-world document/poster images, we reuse the same configuration as in the chart settings, without introducing any additional tools.
> - **Results:** As shown in the table below, PixelCraft consistently outperforms baselines across different backbones (GPT-4.1-mini and GPT-4o).
>
> | Method            | InfoVQA (GPT-4.1-mini) | InfoVQA (GPT-4o) |
> |-------------------|-------------------------|-------------------|
> | CoT               | 77.3                    | 69.9              |
> | Debate            | 74.0                    | 67.6              |
> | Reconcile         | 74.7                    | 67.3              |
> | PixelCraft (Ours) | **79.8**                | **71.6**          |
>
> These results confirm that PixelCraft effectively generalizes to noisy, real-world document images (e.g., infographics, complex posters). Furthermore, we have included qualitative case studies in the revised Appendix F, illustrating how PixelCraft is successfully applied to these real-world document images to resolve complex queries.
>
> > **Response to Question: Confusion regarding Figure 4 (Tool Usage vs. Gain)**
>
> We thank the reviewer for this keen observation. The "Performance Gain" in Figure 4 is calculated as the **conditional accuracy improvement** on the specific subset of queries where the tool was actually activated, rather than an average over the entire dataset.
>
> The discrepancy between "low usage" and "high gain" reflects the specialized nature of these tools:
>
> 1. **Targeted Activation:** Tools like **Legend Masking** and **Adding Auxiliary Lines** are specialized interventions designed to handle specific, high-difficulty visual challenges (e.g., distinguishing confusingly colored lines or aligning distant data points). However, such queries are **relatively rare** in the current benchmark datasets (e.g., CharXiv, ChartQAPro). Consequently, the Dispatcher activates these specialized tools only for a small, high-impact subset where standard Chain-of-Thought (CoT) reasoning often fails.
> 2. **High Impact:** While these tools are called less frequently, their activation is critical for these hard cases. When used, they effectively convert incorrect answers to correct ones, resulting in a substantial performance jump for that specific query subset.
>
> We have clarified this calculation method and interpretation in the revised caption of Figure 4 to avoid future confusion.

---

> ### Author Response · Authors · 2025-11-27
>
> Dear Reviewer yBc8:
>
> Thank you again for your thoughtful feedback and for taking the time to review our work.
>
> We hope that our responses have addressed your concerns. As the rebuttal deadline is approaching, we are writing to kindly check whether our replies have resolved your questions or if any points would benefit from further discussion.
>
> Please feel free to let us know if any further questions arise.

---

### Official Review · Reviewer_k4VN · 2025-11-02

**Soundness:** 3
**Presentation:** 3
**Contribution:** 3
**Rating:** 6
**Confidence:** 4

**Summary:**

This paper aims at solving visual reasoning on structured images, eg. Charts and geometric diagrams. The authors propose PixelCraft, a multi-agent system comprising of three main componets: query-aware agent selection, agent discussion, and iterative self-correction with a planner managed image memory. To achieve accurate visual grounding and image editing, this work also finetunes Qwen2.5-VL-3B into a visual grounding model based on a synthesized training set, and uses it as a reliable Tool Agent. Experiments on multiple recent datasets demonstrated the superior performance of PixelCraft against competitors.

**Strengths:**

- Visual reasoning on structured images is important and has a wide range of applications. This paper proposes an effective multi-agent system for performing visual reasoning on structured images, achieving strong performance across multiple datasets.
- This paper identifies visual grounding as a key obstacle in performing visual reasoning tasks on structured images, and proposes an effective approach to constructing a high-precision visual grounding model based on synthetic training data and MLLM fine-tuning.
- This paper provides detailed experimental results that sufficiently demonstrate the effectiveness of each proposed component.

**Weaknesses:**

- It appears that in the current implementation and evaluation, PixelCraft requires pre-specified tool sets for different task types, such as chart reasoning and geometric reasoning, which limits the assessment of PixelCraft’s generalization capability.
- The paper does not report the computational cost of PixelCraft, particularly in comparison with direct answering and other test-time scaling methods such as Chain-of-Thought (CoT) and Multi-Agent Systems. Comparing both accuracy and computational cost would enable a more comprehensive evaluation of the proposed method’s advantages.

**Questions:**

See Weaknesses.

---

> ### Author Response · Authors · 2025-11-21
> **Response to Reviewer k4VN**
>
> Dear Reviewer k4VN,
>
> We thank the reviewer for the constructive feedback and for recognizing the effectiveness of our visual grounding approach and the comprehensive ablation studies. We address the concerns regarding generalization and computational cost below.
>
> > **Response to W1: Pre-specified tool sets and generalization capability**
>
> **A1**: We thank the reviewer for this insightful comment regarding generalization. We acknowledge that the toolset in the main paper was instantiated for charts and geometry. However, we clarify that these tools were derived via a **semi-automated, data-driven pipeline** (Appendix B) rather than manual crafting, and our system shows strong generalization capability in two aspects:
>
> - **Direct Generalization to Real-World Domains:** To explicitly assess the generalization capability of our *current* toolset, we evaluated PixelCraft on **InfographicsVQA**, a challenging real-world document-understanding benchmark. Crucially, we reused the **exact same tool configuration** as in the chart setting without introducing any additional tools.
> As shown in the table below, PixelCraft consistently outperforms CoT and other agentic baselines (e.g., **+2.5%** over CoT on GPT-4.1-mini).
>
>     | Method            | InfoVQA (GPT-4.1-mini) | InfoVQA (GPT-4o) |
>     |-------------------|-------------------------|-------------------|
>     | CoT               | 77.3                    | 69.9              |
>     | Debate            | 74.0                    | 67.6              |
>     | Reconcile         | 74.7                    | 67.3              |
>     | PixelCraft (Ours) | **79.8**                | **71.6**          |
>
>     This empirical success demonstrates that our high-fidelity tools possess strong cross-domain utility. The generalization stems from shared structural properties across these domains: for instance, Subfigure Cropping allows the model to isolate specific regions in complex infographics just as it does for sub-charts, and Masking Data with Legend effectively removes question-irrelevant visual parts in noisy document layouts. This confirms that our tools capture fundamental structured image reasoning primitives rather than dataset-specific heuristics.
>
> - **Scalable Extension Pipeline**: We further address the limitation regarding flexibility. For domains requiring fundamentally distinct capabilities (e.g., medical imaging), one does not need to rely on heavy manual engineering. Instead, our "Generation–Clustering–Refinement" pipeline (**Appendix B**) serves as a generalizable mechanism to overcome the "pre-specified" limitation. It can be run to automatically discover and semi-automatically refine domain-specific primitives (e.g., `segment_lesion`, `measure_diameter`, or `locate_organ_region`). While we acknowledge that a final round of expert verification is used to ensure precision, this data-driven pipeline significantly reduces the manual effort and reliance on expert knowledge traditionally required to design tool sets, making the system highly scalable to new tasks.
>
> > **Response to W2: Computational cost analysis**
>
> **A2**: Please refer to the general response for further details. In Appendix D.4 of the original manuscript, which is merged into **Appendix E** in the revised version, we compare the latency of PixelCraft to simple CoT prompting. We also report results against Visual Tool Baselines and Multi-Agent Baselines, as described in the general response. In summary, PixelCraft optimizes the balance between cost and performance, achieving substantial accuracy gains over prior systems while maintaining competitive efficiency among agentic frameworks.

---

> ### Author Response · Authors · 2025-11-27
>
> Dear Reviewer k4VN:
>
> Thank you again for your thoughtful feedback and for taking the time to review our work.
>
> We hope that our responses have addressed your concerns. As the rebuttal deadline is approaching, we are writing to kindly check whether our replies have resolved your questions or if any points would benefit from further discussion.
>
> Please feel free to let us know if any further questions arise.

---

### Author Response · Authors · 2025-11-21
**Global Response**

We thank all reviewers for their thoughtful feedback and careful evaluation of our work. We appreciate that you found visual reasoning on structured images to be an important and timely problem (k4VN, hJXi), that our multi-agent system with high-fidelity tools is both novel and technically sound (yBc8,tt9L), that our experiments are comprehensive and convincing across diverse benchmarks and strong agentic baselines (k4VN, yBc8, hJXi), and that the paper is well written and clearly organized (yBc8,tt9L,hJXi). Taking into account all comments, we have revised the draft and provided point-by-point responses to each reviewer, while also addressing common concerns in the general response below.

Several reviewers asked for cost analysis. In the following, we first summarize the main changes since the original submission and then analyze our costs regarding latency and API calls.

### **Revisions**

- Computational cost analysis: We additionally compare the computational cost of PixelCraft with Visual Tool Baselines (Refocus and Visual CoT) and Multi-Agent Baselines (Debate and Reconcile) in **Appendix E**, showing that **PixelCraft is not only effective but also computationally efficient relative to baseline agentic frameworks**. @k4VN, @yBc8, @tt9L, @hJXi
- Evaluation on a real-world document benchmark: We additionally evaluate PixelCraft on InfographicsVQA (see **Appendix F**) without introducing any extra tools, demonstrating strong generalization to noisy, real-world document/poster images. @yBc8, @k4VN, @hJXi

### **Computational cost analysis**

There is a general tradeoff that agentic frameworks achieve better performance by incurring higher computational costs with their iterative planning and tool execution compared to the basic CoT. To demonstrate PixelCraft's relative efficiency compared with other agentic frameworks, we conducted a cost analysis comparing it with both Visual Tool Baselines (Refocus, Visual CoT) and Multi-Agent Baselines (Debate, Reconcile).

We report the average inference latency (seconds per query) and the average number of API calls below. Please note that "Visual CoT" here refers to a linear agent baseline equipped with visual tools but simply includes all the historical images for reasoning.

|Method|Latency (s)|API Calls|Accuracy on CharXiv|
|-|-|-|-|
|Refocus|14.31|2.0|60.7|
| Debate |27.65| 7.0 | 62.4 |
| Reconcile | 25.90| 6.0 | 63.5|
| Visual CoT | 19.31 | 4.4 | 65.0 |
| PixelCraft (Ours) | 16.45| 5.7 | **68.1** |

### **Analysis of Efficiency and Performance:**

- **Scalability vs. Multi-Agent Systems:** As shown in the table, PixelCraft demonstrates superior efficiency compared to existing multi-agent frameworks, reducing latency by approximately **36% to 40%** relative to Reconcile and Debate. This efficiency stems from PixelCraft's **flexible reasoning paradigm**. Unlike Debate or Reconcile, which typically enforce a fixed workflow (e.g., fixed debate rounds) regardless of query complexity, our system employs an adaptive workflow. For simpler queries, the Planner can terminate the process early with fewer steps, whereas complex queries trigger deeper reasoning. This adaptability avoids the computational overhead inherent in rigid multi-agent interactions.
- **Comparison with Linear Agent (Visual CoT):** Crucially, PixelCraft achieves lower latency than the Visual CoT baseline (16.45s compared to 19.31s). This result validates the computational advantage of our **Image Memory** design. Unlike the linear Visual CoT approach, which indiscriminately inputs the entire history of generated images into the context window and thereby increases token count and processing time, our planner selectively recalls only the necessary visual clues. This mechanism ensures a more compact context and faster inference without sacrificing reasoning depth.
- **Trade-off Analysis vs. Refocus:** While PixelCraft incurs a slight increase in latency compared to Refocus, it yields a substantial performance gain. Specifically, on CharXiv using GPT-4.1-mini, PixelCraft achieves an **accuracy improvement of 7.4%** over Refocus.
- **Analyze on API calls**: In PixelCraft, we reduce both context length and the planner’s decision burden by introducing a dispatcher, image memory, critics, and an isolated visual reasoner instead of embedding reasoning logic directly inside the planner. This design yields more API calls overall but much shorter contexts per call, which in turn lowers latency. As shown in the table, PixelCraft issues a moderate number of API calls yet attains superior latency efficiency compared to baseline systems.

In summary, PixelCraft optimizes the balance between cost and performance, delivering substantial accuracy improvements over baselines while maintaining a computational footprint significantly smaller than baseline multi-agent systems. We have incorporated this detailed cost analysis into the revised Appendix E.

---

### Author Response · Authors · 2025-12-02
**Summary of Rebuttal to AC**

Dear Area Chair,

We sincerely appreciate the reviewers' time and constructive feedback. This report summarizes our interactions during the rebuttal, detailing the significant steps taken to resolve concerns.

---
### Work Summary
PixelCraft introduces a **high-fidelity multi-agent system** for structured image reasoning. By synergizing **pixel-level grounding and image processing** with a **dynamic agentic workflow**, our approach enables **high-fidelity and flexible vision reasoning**, addressing a long-standing challenge in achieving faithful reasoning over structured images.
### Key Achievements and Review Summary
- **Recognized Strengths:** Reviewers highlighted that visual reasoning on structured images is an **important and timely problem** (k4VN, hJXi). They found our multi-agent system with high-fidelity tools to be both **novel and technically sound** (yBc8, tt9L). Additionally, the experimental evaluation was praised as **comprehensive and convincing** across diverse benchmarks and strong agentic baselines (k4VN, yBc8, hJXi), and the paper was commended for being **well-written and clearly organized** (yBc8, tt9L, hJXi).

- **General Concerns:** Initial reviews mainly focused on (1) **computational cost/latency**, (2) **generalization capabilities** due to pre-specified tools.

- **Rebuttal Actions:** We conducted **new cost analysis experiments** showing that PixelCraft is **not only effective but also computationally efficient** relative to baseline agentic frameworks, extended evaluation to **real-world document benchmarks (InfographicsVQA)** demonstrating **strong generalization** to noisy images without new tools, added **new comparative baselines (Self-Refine)** verifying our superiority over **test-time scaling** methods under matched compute, and clarified our **novel non-linear reasoning** and **dynamic agentic workflow** contributions.
---
### Detailed Response to Reviewers
1. Reviewer k4VN (6 – Marginally Above Threshold)
- **Initial Assessment:** Praised visual grounding effectiveness but raised concerns about **generalization** and requested **computational cost analysis**.

- **Rebuttal:**
    - **Generalization:** Added experiments on **InfographicsVQA** (Appendix F), demonstrating PixelCraft generalizes to noisy, real-world documents (**+2.5% over CoT**) without adding new tools. Clarified our **semi-automated tool generation pipeline** (Appendix B).
    - **Cost:** Provided detailed **latency and API call analysis** (Appendix E), showing PixelCraft is **efficient** relative to other agentic frameworks.

2. Reviewer yBc8 (6 – Marginally Above Threshold)
- **Initial Assessment:** Acknowledged originality but questioned **scalability/cost** and reliance on specialized benchmarks rather than more realistic and noisy images like document.

- **Rebuttal:**
    - **Scalability:** Demonstrated that PixelCraft reduces latency by **36-40%** compared to Multi-Agent baselines (Debate, Reconcile) while achieving higher accuracy.
    - **Benchmarks:** The addition of **InfographicsVQA** demonstrates the generalization of PixelCraft to more realistic and noisy images.

3. Reviewer tt9L (8 – Accept)
- **Initial Assessment:** Strong endorsement but requested **cost comparison** against vanilla Visual CoT and **Self-Refine/Critic baselines**.

- **Rebuttal:**
    - **Cost:** Shown to be **faster than linear Visual CoT** (16.45s vs 19.31s) due to our **efficient Image Memory** design.
    - **New Baseline:** Implemented a **Self-Refine baseline** (matched compute). PixelCraft significantly outperforms it (**68.1% vs 62.0%**), proving the architecture's value over simple test-time scaling.

4. Reviewer hJXi (4 – Did not participate in discussion)
- **Initial Assessment:** Praised the **challenging problem addressed**, **comprehensive experimental setup** against advanced baselines, and **high-quality presentation** but criticized **efficiency**, **novelty of the agentic framework**, and **fixed toolset**.

- **Rebuttal:**
    - **Efficiency:** Provided evidence that our **adaptive workflow** is computationally efficient compared to standard multi-agent systems.
    - **Novelty:** Clarified 1) the shift from **rigid/linear execution** to **non-linear visual reasoning** via selective memory recall—a fundamental structural improvement; 2) the improvement from **rigid multi-agent workflow** to **dynamic agentic workflow** by decomposing the workflow into atomic agent actions and then agentically calling them.
    - **Tool Contributions:** Refuted "fixed toolset" limitation via our **data-driven generation pipeline**, which provides higher-fidelity tools compared with prior works and achieves empirical success on **out-of-domain tasks** (InfoVQA).
---
### Conclusion
In light of the substantial improvements and the positive consensus from the reviewers, we sincerely hope the AC will give our paper the most favorable consideration.

Best regards,

The Authors of Submission 5711

---

### Meta-Review · Area_Chair_Ubhj · 2026-01-05

**Summary:**

This paper introduces a multi-agent system designed for high-fidelity visual reasoning on structured images such as charts and geometric diagrams. By integrating a fine-tuned grounding model with classical CV operators, the system achieves precise pixel-level processing that effectively minimizes perceptual slips. A key innovation is the image memory mechanism that enables a non-linear reasoning workflow, allowing the planner to selectively recall visual clues to facilitate branching and backtracking. PixelCraft establishes new performance standards on challenging benchmarks, including CharXiv and ChartQAPro. Furthermore, the rebuttal successfully demonstrated that the system is more computationally efficient than linear Visual CoT and generalizes effectively to noisy real-world benchmarks like InfographicsVQA without additional tool modification.

**Reviewer Concerns:**

Addressed: Authors resolved efficiency concerns, proving PixelCraft is faster than linear Visual CoT and multi-agent baselines. Generalization was validated through success on InfographicsVQA without tool modifications.

Outstanding: Reviewer hJXi maintains that the agentic novelty is incremental. The dependency on a curated toolset is still perceived as a potential limitation for generalizability across distinct domains.

**Reviewer Scores:**

Reviewer tt9L (8): Recommended acceptance, citing the high-fidelity processing and the architecture's demonstrated value over simple test-time scaling.

Reviewer k4VN (6): Rated the paper as marginally above the acceptance threshold. This reviewer did not provide further comments during the post-rebuttal discussion phase.

Reviewer yBc8 (6): Supported marginal acceptance, noting the system's scalability and effectiveness on noisy, realistic images.

Reviewer hJXi (4): Recommended marginal rejection and did not participate in the post-rebuttal discussion; initial concerns focused on methodological novelty and toolset flexibility.

---

### Decision · Program_Chairs · 2026-01-26

Accept (Poster)